# LOOK IN THE MIRROR: MOLECULAR GRAPH CONTRASTIVE LEARNING WITH LINE GRAPH

## ABSTRACT

Trapped by the label scarcity in molecular property prediction and drug design, graph contrastive learning came forward. A general contrastive model consists of a view generator, view encoder, and contrastive loss, in which the view mainly controls the encoded information underlying input graphs. Leading contrastive learning works show two kinds of view generators, that is, random or learnable data corruption and domain knowledge incorporation. While effective, the two ways also lead to molecular semantics altering and limited generalization capability, respectively. Thus, a decent view that can fully retain molecular semantics and is free from profound domain knowledge is supposed to come forward. To this end, we relate molecular graph contrastive learning with the line graph and propose a novel method termed LGCL. Specifically, by contrasting the given graph with the corresponding line graph, the graph encoder can freely encode the molecular semantics without omission. While considering the information inconsistency and over-smoothing derived from the learning process because of the mismatched pace of message passing in two kinds of graphs, we present a new patch with edge attribute fusion and two local contrastive losses for performance fixing. Compared with state-of-the-art (SOTA) methods for view generation, superior performance on molecular property prediction suggests the effectiveness of line graphs severing as the contrasting views.

## 1 INTRODUCTION

A deep understanding of molecular properties plays a vital role in the chemical and pharmaceutical domains. In order to computationally discover novel materials and drugs, the molecules will be abstractly regarded as graphs, in which atoms are vertices and bonds are edges Gilmer et al. (2017); Goh et al. (2017); Chen et al. (2018a). Thus, the marriage between molecular property prediction and graph learning captured a bunch of researchers and showed their happiness in several fields Yang et al. (2019); Song et al. (2020); Chen et al. (2021); Wu et al. (2022a). However, this relationship faces the challenges of label scarcity, as deep learning methods are known to consume massive amounts of labeled data, and annotated data are often of limited size and hard to acquire when considering many specific domains. In addition, given the immense differentiation among chemical molecules, existing supervised models could be barely reused in unseen cases Hu et al. (2020); Rong et al. (2020). Therefore, there are increasing demands for molecular representation learning in an unsupervised or self-supervised manner.

Plenty of works have attempted to learn molecule representations discarding the supervision of labels, like graph context prediction Liu et al. (2019), graph-level motif prediction Rong et al. (2020) and masked attribute prediction Hu et al. (2020). In light of the contrastive learning from computer vision, researchers go one step further to model molecules in a contrastive manner with data augmentations You et al. (2020); Suresh et al. (2021). Considering the inherent characteristics of chemical molecules, graph contrastive learning incorporating well-designed domain knowledge has also shown excellent capacity in molecular properties prediction Sun et al. (2021); Fang et al. (2022).

Analogously, everything comes with a price. Inspecting the generated views in previous molecular graph contrastive learning unveils two intrinsic limitations. First, data augmentation-based methods adopting random or learnable corruption (e.g., node/edge dropping and graph generation) would lead to inevitable variance in the crucial semantics and further misguide the contrastive learning

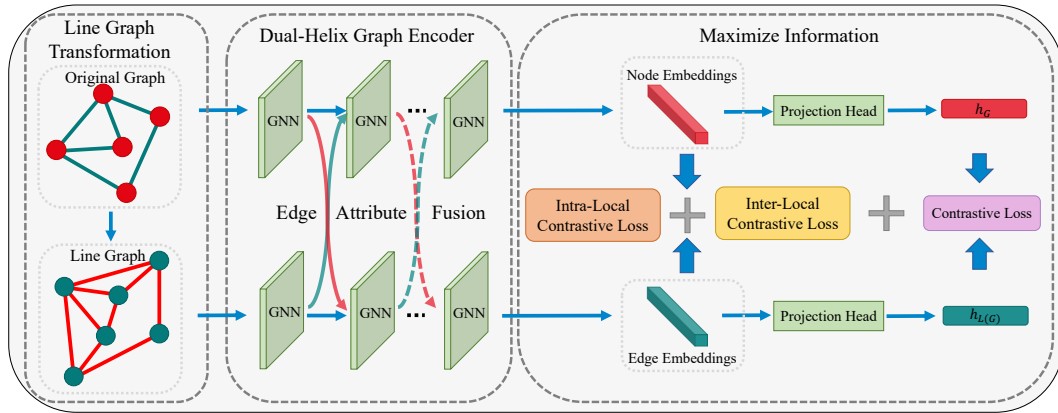

Figure 1: **Framework overview of LGCL.** Contrasted views consist of the original graph and the corresponding line graph. Input graphs are encoded by a dual-helix graph encoder with edge attribute fusion for information consistency. The whole model is jointly optimized via minimizing the NT-Xent loss and the two local contrastive losses.

You et al. (2020); Sun et al. (2021). Second, based on predefined sub-structure substitution rules Sun et al. (2021) or contrasted with 3-dimension geometric views Liu et al. (2022); Stärk et al. (2022), domain knowledge-based methods intend to alleviate the problem of semantic alteration. While effective, they are stinted to the profound domain knowledge that is unfriendly to researchers without such knowledge, thus limiting their generalization capability in other domains.

In this context, we are seeking for a decent view that will not be bothered by prefabricated domain knowledge and can maintain the molecular semantic information integrally. Fortunately, we met the line graph, also known as congruent graph in graph theory Whitney (1932); Harary & Norman (1960); Jung (1966). In a line graph, the nodes correspond to the edges of the original graph, and the edges refer to the common nodes of the pair edges in the original graph. In particular, the isomorphism of two line graphs is judged to be consistent with the corresponding two original graphs Whitney (1932); Jung (1966), which ensures the congruent semantic structure after line graph transformation. In light of the line graph, we propose a method termed LGCL to tackle our expectations.

The framework of LGCL is shown in Figure 1. Specifically, to fill the framework demanding two views, all input molecular graphs are transformed into the corresponding line graph. On such a basis, LGCL equips with a dual-helix graph encoder to learn the hidden representation of two views. Note that, due to the different pace of message passing in the original graph and the corresponding line graph, two issues derive from the learning process, that is, information inconsistency and over-smoothing. For information consistency, we further update the graph encoder with edge attribute fusion to bridge the edge attributes between the two kinds of graphs. Over-smoothing is addressed by a novel intra-local contrastive loss based on the idea of NT-Xent loss; put differently, the intra-local contrastive loss aims to maximize the consense between the edge pairs in the two corresponding views and minimize the consense between different edge pairs within the same views. Moreover, we further give an inter-local contrastive loss to enhance the representation learning.

The effectiveness of LGCL is verified under the ubiquitous setting of transfer learning for molecular property prediction Hu et al. (2020). Through pre-training on two million molecular graphs from ZINC15, LGCL shows superior performance on six out of eight benchmarks for molecular property prediction and acquires the highest position on both average ROC-AUC and average ranking. Additionally, we delve deeper into the proposed components via analytical experiments to further assess their benefits. The contributions are elaborated below:

- To the best of our knowledge, we are the first to figure out a way to freely and fully excavate molecular semantics within graph contrastive learning.

- Inspired by the line graph, we present an approach, termed LGCL, to tackle our expectations, in which edge attribute fusion and two local contrastive losses are united to address the concomitant issues and enhance molecular representation learning.

- Leveraging eight benchmarks for molecular property prediction under the setting of transfer learning, LGCL exhibits its superiority against the SOTA methods for view generation.

## 2 RELATED WORKS

This research focuses on molecular graph contrastive view generation, especially considering the case that is free from the intricate domain knowledge. We will elaborate on these topics below.

**Molecular graph contrastive learning.** Along with the development of graph contrastive learning, plenty of research efforts have been devoted to designing contrastive learning models for molecular graphs Sun et al. (2021); Xu et al. (2021); Fang et al. (2022); Stärk et al. (2022); Liu et al. (2022); Li et al. (2022a). Besides random or learnable corruption, several works presented various contrastive learning models to embed the molecular geometry information by means of contrasting the generic 2D graph with its 3D conformers Liu et al. (2022); Stärk et al. (2022); Li et al. (2022a). They indeed get rid of the semantic altering issue caused by random corruption on molecular graphs, while introducing another semantic altering issue caused by 3D conformers, because a single 2D molecular graph generally has multiple conformers with different chemical properties Stärk et al. (2022). To enhance the performance in molecular property prediction, the domain knowledge-driven contrastive learning frameworks were proposed to preserve the semantics of graphs in the augmentation process Sun et al. (2021); Fang et al. (2022). However, their learning capability heavily relies on the dissolved domain knowledge, that is the well-designed substitution rules in MoCL Sun et al. (2021) and the prefabricated associations among chemical elements in KCL Fang et al. (2022). Furthermore, the domain knowledge varies across domains, which limits the application of these methods.

Recently, besides the contrasting view exploration, GraphLoG Xu et al. (2021) and OEPG Yang & Hong (2022) are built upon the generic graph contrastive learning methods to discover the global semantic structure underlying the whole dataset and present excellent performance. In this work, we are devoted to the domain of contrasting view generation, which is orthogonal to the works for dataset semantic structure exploration; put differently, extensive works for contrasting view generation can work with the framework of GraphLoG and OEPG to produce more superior performance.

**Line graph.** The line graph is a classic concept and has a long history in graph theory Whitney (1932); Harary & Norman (1960); Jung (1966). In a line graph, the nodes correspond to the edges of the original graph, and the edges refer to the common nodes of the pair edges in the original graph. Thus, the graph neural networks (GNNs) built on line graphs are capable of encoding edge features and enhancing feature learning on graphs. Recently, based on the line graph structures, several line graph neural networks have shown promising performance on various graph-related tasks Chen et al. (2018b); Jiang et al. (2019); Bandyopadhyay et al. (2020). In the chemistry domain, the structure of a compound can be treated as a graph, where the edges derived from chemical bonds link the corresponding atom nodes. Thus, the edges in such graphs have different properties and various functions. In generic GNNs, however, message-passing operations among nodes do not pay enough attention to the edge properties. Fortunately, the line graph structure enables generic GNNs to take advantage of the edges as equals as nodes Jiang et al. (2019); Chen et al. (2018b).

To this far, there is still no graph contrastive learning model to encode the molecular semantics integrally without well-designed domain knowledge. In this paper, we revisit the line graph from the angle of graph contrastive learning. Based on it, we design a novel contrastive model, termed LGCL, to freely and fully excavate molecular semantics.

## 3 PRELIMINARIES

Here, we first present some preliminary concepts and notations. In this work, let $\mathbb{G} = \{G_1, G_2, \cdots, G_N\}$ be a graph dataset with size $N$, and a molecular graph can be formulated as $G = (V, E, X_V, X_E)$, where $V$ is the node set, $E$ is the edge set, $X_V \in \mathbb{R}^{|V| \times \mathbb{V}}$ denotes the node features, and $X_E \in \mathbb{R}^{|E| \times \mathbb{E}}$ stands for the edge attributes.

## 3.1 GRAPH REPRESENTATION LEARNING

In generic GNNs, the message-passing scheme is adopted for information transmission among nodes Xu et al. (2019); Wu et al. (2022b). Through stacking $L$ layers, a GNN will produce a hidden representation $h_v \in \mathbb{R}$ with $L$-hop neighbor information for each node and a feature vector $h_G \in \mathbb{R}$ via a global readout function for the entire graph $G$. Each node $v$ is initialized with node feature $X_v$ and sent to the GNN input. Formally, the $l$-th layer of a GNN Xu et al. (2019) can be written as

$$\hat{h}_v^{(l)} = \text{AGGREGATE}^{(l)}(\{h_u^{(l-1)}|u \in N(v)\}), \tag{1}$$

$$h_v^{(l)} = \text{COMBINE}^{(l)}(\hat{h}_v^{(l)}, h_v^{(l-1)}), \tag{2}$$

where $h_v^{(l)}$ represents the feature vector of node $v$ at the $l$-th iteration, $N(v)$ covers the 1-hop neighbors of $v$, AGGREGATE denotes the crucial message-passing scheme in GNNs, and COMBINE is used to update the hidden feature of $v$ via merging information from its neighbors and itself. Finally, a GNN can produce the feature vector $h_G$ of the entire graph with a prefabricated readout function:

$$h_G = \text{READOUT}(\{h_v|v \in \mathcal{V}\}), \tag{3}$$

where READOUT aggregates the final set of node representations.

## 3.2 GRAPH CONTRASTIVE LEARNING

In a generic graph contrastive learning model, two correlated views from the same graph $G_i$ are required for contrasting and generally produced by two augmentation operations. Here, we denote the augmented views as $\tilde{G}_i^1$ and $\tilde{G}_i^2$. Then, a graph encoder and a projection head are stacked behind the two augmentation operators to map the correlated views into an embedding space and yield corresponding feature vectors $h_i^1$ and $h_i^2$. The released hidden representations are supposed to contain the essential features of the original graph $G$ so that they can recognize themselves from the others. Thus, the objective of graph contrastive learning is to maximize the consensus between the two positive views via the widespread NT-Xent loss Chen et al. (2020):

$$\mathcal{L}_i = -\log \frac{e^{sim(h_i^1, h_i^2)/\tau}}{\sum_{j=1, j \neq i}^{N} e^{sim(h_i^1, h_j^2)/\tau}}, \tag{4}$$

where $N$ is the batch size, $\tau$ refers to the temperature parameter, and $sim(h^1, h^2)$ denotes a cosine similarity function $\frac{h^{1\top}h^2}{||h^1|| \cdot ||h^2||}$. The numerator part is the similarity of the correlated views as positive pair. The rest pairs that consist of views from different graphs are regarded as negative pairs and act as the denominator part. Note that the negative pairs can come from two directions, put differently, $h_i^1$ can pair with all $h_j^2$, and $h_i^2$ can pair with all $h_j^1$.

## 4 METHODOLOGY

In this section, we bring about the proposed graph contrastive learning framework, termed *LGCL*, by revisiting the line graph of corresponding molecules. Given the issue of label scarcity in real-world graph data, LGCL is designed to encode the molecular semantics integrally and free from well-designed domain knowledge. Specifically, to produce two contrastive views without any loss of molecular semantics, we first need to transform the given molecule to the corresponding line graph. On such a basis, LGCL equips with a dual-helix graph encoder to learn the hidden representation of two views with edge attribute fusion. In particular, besides the ubiquitous contrastive loss for the readout graph representations, we further propose two local contrastive losses to enhance representation learning and alleviate the over-smoothing issue in deep GNNs. Next, we elaborate on the LGCL framework below.

## 4.1 LINE GRAPH TRANSFORMATION

Here, we first present an illustration of line graph transformation from a simple graph. As shown in Figure 2, let $G = (V, E)$ be a simple undirected graph, the output line graph $L(G)$ after transformation is such a graph that reveals the adjacencies of edges in $G$. Specifically, each edge in $G$

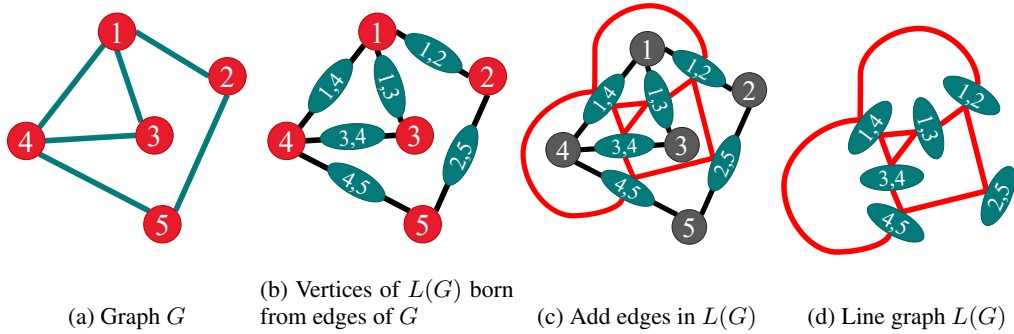

(a) Graph $G$    (b) Vertices of $L(G)$ born from edges of $G$    (c) Add edges in $L(G)$    (d) Line graph $L(G)$

Figure 2: **An illustration of line graph transformation.** (a) shows a simple undirected graph $G$; (b) reveals the derivation of vertices in line graph, every vertex of line graph is marked with green and labeled with the pair nodes of the corresponding edge in $G$; (c) establishes the associations in $L(G)$ based on the common nodes of two edges; (d) delivers the output line graph $L(G)$ of the original graph $G$.

is mapped into a node of $L(G)$ and each edge in $L(G)$ indicates that the corresponding two vertices have a common node in $G$. Formally, the line graph can be written as $L(G) = (V_L, E_L)$, where $V_L = \{(v_i, v_j)|(v_i, v_j) \in E)\}$ and $E_L = \{((v_i, v_j), (v_j, v_k))|\{(v_i, v_j), (v_j, v_k)\} \subset E\}$. At this point, we have settled the topology transformation of line graphs. Besides the relationships among nodes and edges, the node and edge attributes underlying the molecular graph should also be delivered to the corresponding line graph. In this paper, based on the one-to-one correspondence between the edges of $G$ and the nodes of $L(G)$, the node attributes of the line graph can be directly obtained from the edge attributes of original graphs, that is, $X_{V_L} = X_E$. As for the edge attributes of $L(G)$, because several edges in the line graph would correspond to the same node in the original graph, a mapping function with such relationships is required to endow the line graph edge with the original node attribute. In the light of $E_L$ after the line graph transformation, the edge attribute mapping function can be formulated as $M(e_L) = (v_i, v_j) \cap (v_j, v_k)$, thus the edge attributes of the line graph can be obtained via $X_{E_L} = MX_V$. Finally, the line graph of a molecular graph is given by $L(G) = (V_L, E_L, X_{V_L}, X_{E_L})$. According to Roussopoulos's algorithm Roussopoulos (1973), the time complexity of line graph transformation is $O(\max(|V|, |E|))$.

As stated in the Whitney graph isomorphism theorem Whitney (1932), the isomorphism of two line graphs is judged to be consistent with the corresponding two original graphs, which convinces us that the semantic structure information of $G$ is encoded in the line graph $L(G)$. In particular, as described in the line graph transformation, there is a one-to-one correspondence between the edges in the graph $G$ and the vertices in the line graph $L(G)$. Therefore, a vertex with $e$ edges in $G$ will produce $e \times (e-1)/2$ edges in $L(G)$. Meanwhile, the message-passing frequency around this node will drift from $O(e)$ in $G$ to $O(e^2)$ in $L(G)$, put differently, this node feature in $G$ is only passed to $e$ neighbors, while the corresponding line graph will pass such information to $e \times (e-1)/2$ nodes. Actually, in light of the line graph capacity in Table A.1, the line graph encoder only requires about 50% more computation than the original graph encoder. While this nature of the line graph could cause two inevitable issues in the contrastive learning framework with stacked graph convolutional layers, that is information inconsistency and over-smoothing. In this paper, we propose two approaches, edge attribute fusion and two local contrastive losses, to alleviate the two issues and strengthen molecular representation learning. Next, we give a detailed description.

## 4.2 EDGE ATTRIBUTE FUSION

In the chemistry domain, the structure of a compound can be treated as a graph, where the edges derived from the chemical bonds link the corresponding atom nodes. Thus, the edges in such graphs have different properties and various functions. Besides the topology information weaved by atoms, a well-designed graph convolution with edge attributes plays a crucial role in molecular property and protein function prediction.

Given a molecular graph, its input node features and edge features are both represented as a 2-dimensional categorical vector (see Appendix A for details), denoted as $X_V \in \mathbb{R}^{|V| \times 2}$ and $X_E \in \mathbb{R}^{|E| \times 2}$, respectively. In previous works regarding molecular property prediction Hu et al. (2020), the raw node categorical vectors are embedded in the input layer by

$$h_v^{(0)} = \text{EMBEDDING}(x_v^0) + \text{EMBEDDING}(x_v^1), \tag{5}$$

where $x_v^0$ and $x_v^1$ are the atomic number and chirality tag of node $v$, respectively. EMBEDDING$(\cdot)$ denotes an embedding function that transfers a single integer into a $d$-dimensional vector space. Meanwhile, the raw edge categorical vectors are embedded in each layer by

$$h_e^{(l)} = \text{EMBEDDING}(x_e^0) + \text{EMBEDDING}(x_e^1), \tag{6}$$

where $x_e^0$ and $x_e^1$ represent the bond type and bond direction, respectively, and $l$ denotes the index of GNN layers. At the $l$-th layer, the node representation can be updated by

$$h_v^{(l)} = \sigma(\text{MLP}^{(l)}(h_v^{(l-1)} + \sum_{u \in N(v)} h_u^{(l-1)} + \sum_{e \in \{(v,u)|u \in N(v) \cup \{v\}\}} h_e^{(l-1)})), \tag{7}$$

where $\sigma(\cdot)$ is an activation function, and $(v, v)$ represents the self-loop edge.

Under this GNN architecture, the output molecular representations will be decorated with edge attributes. However, as discussed above, there is a significant difference in message-passing frequency between the original graph and the corresponding line graph, which could lead to information inconsistency between the outputs. Here, we present a novel *edge attribute fusion* approach to tackle this issue. Specifically, we bridge the edge information between the molecular graph and line graph to help the original graph encoder keep pace with the line graph encoder. The edge and node embeddings are still employed as the initial edge attributes in the first layer (i.e., $l = 0$). As for $l \geq 1$, the edge attributes of the original graph are obtained from the node hidden features in the line graph, which is formally given by

$$h_{G \cdot (v_i, v_j)}^{(l)} = h_{L(G) \cdot (v_i, v_j)}^{(l-1)}, \tag{8}$$

where $(v_i, v_j) \in E$ and $(v_i, v_j) \in V_L$. Correspondingly, the edge attributes of the line graph can be updated by the node hidden features in the original graph, which is formally formulated as:

$$h_{L(G) \cdot ((v_i, v_j), (v_j, v_k))}^{(l)} = h_{G \cdot v_j}^{(l-1)}, \tag{9}$$

where $(v_i, v_j) \in E$, $(v_j, v_k) \in E$ and $((v_i, v_j), (v_j, v_k)) \in E_L$. Based on the dual-helix graph encoder with edge attribute fusion, the hidden features from the line graph will be dissolved into the original graph representations, allowing information consistency between the two contrastive views and enhancing molecular representation learning.

## 4.3 Intra-Local Contrastive Loss

In this part, we look forward to tackling the over-smoothing issue introduced by the line graph. Motivated by NT-Xent loss for contrastive learning, an intra-local contrastive loss is proposed. The design concept of NT-Xent loss aims to maximize the representation similarities of positive pairs consisting of hidden features of the same molecules and enforce dissimilarity of negative pairs comprising hidden features of different molecules simultaneously. Similarly, the proposed intra-local contrastive loss seeks to optimize the consensus between the same nodes as opposed to different nodes in a single graph. Considering the one-to-one correspondence between the edges in $G$ and the vertices in $L(G)$, the contrastive samples of this loss are composed of the edge hidden features in $G$ and the node hidden features in $L(G)$. Thus, given a graph $G$, the intra-local contrastive loss of one edge pair is formally defined as:

$$\mathcal{L}_{IntraC}^{e_i} = -\log \frac{e^{sim(\tilde{h}_{G \cdot e_i}, h_{L(G) \cdot e_i})/\tau}}{\sum_{j=1, j \neq i}^{|E|} e^{sim(\tilde{h}_{G \cdot e_i}, h_{L(G) \cdot e_j})/\tau}}, \tag{10}$$

where $e_i = (v_m, v_n)$, $e_i \in E$ and $(v_m, v_n) \in V_L$. In particular, the edge representations from $G$ are formed by such hidden features of the two endpoints of each edge, that is, $\tilde{h}_{G \cdot e_i} = \text{MLP}([h_{v_m}, h_{v_n}])$. In light of this contrastive loss designed inside the graph, we look forward to reducing the similarity between different nodes and further alleviating the over-smoothing.

### 4.4 INTER-LOCAL CONTRASTIVE LOSS

Here, we give another contrastive loss to enhance molecular graph contrastive learning. As we are capable of enforcing dissimilarity between different edge representations, we move forward to generalizing the edge dissimilarity to all contrasted samples. Our critical insight is that the widespread NT-Xent loss only provides graph-level contrast, while a contrastive angle based on node representations would also be meaningful in crucial structure identification. In light of the intra-local contrastive loss, the inter-local contrastive loss can be formally formulated as:

$$\mathcal{L}_{InterC}^{e_i} = -\log \frac{e^{sim(\tilde{h}_{G \cdot e_i}, h_{L(G) \cdot e_i})/\tau}}{\sum_{\hat{G} \neq G}^{\mathbb{G}} \sum_{j=1}^{|E_{\hat{G}}|} e^{sim(\tilde{h}_{G \cdot e_i}, h_{L(\hat{G}) \cdot e_j})/\tau}},$$ (11)

where $e_i \in E_G$, $e_j \in V_{L(\hat{G})}$ and $\mathbb{G}$ represents a training batch. Note that the negative pairs of the inter-local contrastive loss also come from two directions.

Currently, we have presented the main components of the proposed LGCL that aims to help molecular graph contrastive learning free from well-designed domain knowledge and maintain the semantics. For unsupervised molecular graph representation learning, the final objective function of LGCL for pre-training is given by

$$\min \mathcal{L} = \mathcal{L}_G + \alpha \mathcal{L}_{InterC} + \beta \mathcal{L}_{IntraC},$$ (12)

where $\mathcal{L}_G$ denotes the NT-Xent loss, $\alpha$ and $\beta$ are two hyper-parameters for loss weight controlling.

## 5 EXPERIMENT

In this section, we are devoted to evaluating LGCL with extensive experiments [1]. Following the procedure of pre-training and fine-tuning, we validate the effectiveness of our approach against SOTA competitors for view generation. Furthermore, we carry out analytical studies to assess each proposed component. Unsupervised and semi-supervised learning are shown in appendix.

### 5.1 EXPERIMENTAL SETUP

To be in line with the previous graph contrastive learning methods without prefabricated domain knowledge and make the comparisons fair, we follow the experimental setup under the guidance of Hu et al. (2020).

**Pre-training dataset.** ZINC15 Sterling & Irwin (2015) dataset is adopted for LGCL pre-training. In particular, a subset with two million unlabeled molecular graphs are sampled from the ZINC15.

**Pre-training details.** In the graph encoder setting in Hu et al. (2020), a Graph Isomorphism Network (GIN) Xu et al. (2019) with five convolutional layers is adopted for message passing. In particular, the hidden dimension is fixed to 300 across all layers and a pooling readout function that averages graph nodes is hired for NT-Xent loss calculation with the scale parameter $\tau = 0.1$. The hidden representations at the last layer are injected into the average pooling function. An Adam optimizer Kingma & Ba (2015) is employed to minimize the integrated losses produced by the 5-layer GIN encoder. The batch size is set as 256, and all training processes will run 100 epochs. The two hyper-parameters (i.e., $\alpha$ and $\beta$) for loss weight controlling are both set as 1.

**Fine-tuning dataset.** We employ the eight ubiquitous benchmarks from the MoleculeNet dataset Wu et al. (2018) to validate LGCL as downstream experiments. These benchmarks include a variety of molecular tasks like physical chemistry, quantum mechanics, physiology, and biophysics. For dataset split, the scaffold split scheme Chen et al. (2012) is adopted for train/validation/test set generation. Table A.1 summarizes the basic characteristics of the datasets, such as the size, tasks and molecule statistics. Detailed descriptions can be found in Appendix A.

**Fine-tuning details.** For downstream tasks, a linear layer is stacked after the pre-trained graph encoders for final property prediction. The downstream model still employs the Adam optimizer for 100 epochs fine-tuning. All experiments on each dataset are performed for ten runs with different

---

[1]The code of LGCL will be public after acceptance.

seeds, and the results are the averaged ROC-AUC scores (%) $\pm$ standard deviations. The hyper-parameters tuned for each dataset are: (a) the learning rate $\in \{0.01, 0.001, 0.0001\}$; (b) the batch size $\in \{32, 128\}$; (c) the dropout ratio $\in \{0, 0.5\}$. The node representations for graph pooling are adopted from the last layer or the concatenation of all layers. These hyper-parameters are selected by the grid search on the validation sets.

**Baselines.** In this paper, we choose the SOTA competitors that follows the experimental setup in Hu et al. (2020). The first category is self-supervised graph learning algorithms, including EdgePred, AttrMsking, ContexPred Hu et al. (2020), Infomax Velickovic et al. (2019), and GraphMAE Hou et al. (2022). The second category are graph contrastive learning methods for view generation, such as GraphCL You et al. (2020), JOAO(v2) You et al. (2021), LP-Info You et al. (2022), AutoGCL Yin et al. (2022), GraphMVP Liu et al. (2022), RGCL Li et al. (2022b) and D-SLA Kim et al. (2022).

## 5.2 RESULTS

The results of LGCL along with SOTA competitors for molecular property prediction on eight benchmarks are shown in Table 1. To summarize, the proposed graph contrastive learning framework with the line graph, LGCL, obtains superior performance compared with the previous works. Specifically, in light of the last column for average rank, our method seizes the highest ranking position from SOTA contrastive learning methods as well as self-supervised learning methods, and a significant ranking improvement can be witnessed as opposed to the second place (D-SLA gives the A.R. 5.0). In particular, LGCL achieves the best performance on six out of eight benchmarks, and the best comprehensive performance is also with us (see the penultimate column). Thus, we can conclude that LGCL captures the molecular semantic information well in the absence of well-designed domain knowledge, and the line graph provides an excellent contrastive view without altering the molecular semantics.

Table 1: Average test ROC-AUC (%) $\pm$ Std. over different 10 runs of LGCL along with all baselines on eight downstream molecular property prediction benchmarks. The results of baselines are derived from the published works. **Bold** indicates the best performance among all baselines. Avg. shows the average ROC-AUC over all datasets. A.R. denotes the average rank.

| Dataset | BBBP | Tox21 | ToxCast | SIDER | ClinTox | MUV | HIV | BACE | Avg. | A.R. |
|---|---|---|---|---|---|---|---|---|---|---|
| No Pre-Train | 65.8±4.5 | 74.0±0.8 | 63.4±0.6 | 57.3±1.6 | 58.0±4.4 | 71.8±2.5 | 75.3±1.9 | 70.1±5.4 | 66.96 | 13.5 |
| Infomax | 68.8±0.8 | 75.3±0.6 | 62.7±0.4 | 58.4±0.8 | 69.9±3.0 | 75.3±2.5 | 76.0±0.7 | 75.9±1.6 | 70.29 | 11.2 |
| EdgePred | 67.3±2.4 | 76.0±0.6 | 64.1±0.6 | 60.4±0.7 | 64.1±3.7 | 74.1±2.1 | 76.3±1.0 | 79.9±0.9 | 70.28 | 9.3 |
| AttrMasking | 64.3±2.8 | 76.7±0.4 | 64.2±0.5 | 61.0±0.7 | 71.8±4.1 | 74.7±1.4 | 77.2±1.1 | 79.3±1.6 | 69.90 | 8.9 |
| ContextPred | 68.0±2.0 | 75.7±0.7 | 63.9±0.6 | 60.9±0.6 | 65.9±3.8 | 75.8±1.7 | 77.3±1.0 | 79.6±1.2 | 70.89 | 8.1 |
| GraphMAE | 72.0±0.6 | 75.5±0.6 | 64.1±0.3 | 60.3±1.1 | 82.3±1.2 | 76.3±2.4 | 77.2±1.0 | 83.1±0.9 | 73.85 | 5.8 |
| GraphCL | 69.68±0.67 | 73.87±0.66 | 62.40±0.57 | 60.53±0.88 | 75.99±2.65 | 69.80±2.66 | 78.47±1.22 | 75.38±1.44 | 70.77 | 10.9 |
| JOAO(v2) | 71.39±0.92 | 74.27±0.62 | 63.16±0.45 | 60.49±0.74 | 80.97±1.64 | 73.67±1.00 | 77.51±1.17 | 75.49±1.27 | 72.12 | 9.0 |
| LP-Info | 71.40±0.55 | 74.54±0.45 | 63.04±0.30 | 59.70±0.43 | 74.81±2.73 | 72.99±2.28 | 76.96±1.10 | 80.21±1.36 | 71.71 | 9.9 |
| AD-GCL | 70.01±1.07 | 76.54±0.82 | 63.07±0.72 | 63.28±0.79 | 79.78±3.52 | 72.30±1.61 | 78.28±0.97 | 78.51±0.80 | 72.72 | 7.1 |
| AutoGCL | **73.36±0.77** | 75.69±0.29 | 63.47±0.38 | 62.51±0.63 | 80.99±3.38 | 75.83±1.30 | 78.35±0.64 | 83.26±1.13 | 74.18 | 4.1 |
| GraphMVP | 68.5±0.2 | 74.5±0.4 | 62.7±0.1 | 62.3±1.6 | 79.0±2.5 | 75.0±1.4 | 74.8±1.4 | 76.8±1.1 | 71.70 | 9.9 |
| RGCL | 71.42±0.66 | 75.20±0.34 | 63.33±0.17 | 61.38±0.61 | **83.38±0.91** | 76.66±0.99 | 77.90±0.80 | 76.03±0.77 | 73.16 | 5.9 |
| D-SLA | 72.60±0.79 | 76.81±0.52 | 64.24±0.50 | 60.22±1.13 | 80.17±1.50 | 76.64±0.91 | 78.59±0.44 | 83.81±1.01 | 74.14 | 3.8 |
| LGCL | 70.99±1.05 | **76.95±0.43** | **64.71±0.72** | **63.37±0.56** | 77.59±1.54 | **77.70±3.00** | **78.69±1.10** | **84.68±0.73** | **74.33** | **2.6** |

## 5.3 ABLATION STUDY

Here, we delve deeper into the performance influence of each proposed component. First, we analyze the performance boosting from the introduction of the line graph and edge attribute fusion without ZINC15 pre-training. As for the two local contrastive losses, we present the test results of various combinations from these parts following the transfer learning settings. The detailed discussions are as follows.

**The effect of line graph.** In Figure 3, we analyze the effect of the line graph. In comparison with the red bar (i.e., 'No Pre-Train') that denotes the results from random initialization, introducing the line graph (i.e., 'No Pre-Train w/LG') shows an overall superior performance, which empirically suggests that the semantics underlying the edges can be better captured by the line graph.

**The effect of edge attribute fusion.** Based on the performance boosting of the line graph, we further present the results with edge attribute fusion in Figure 3 (i.e., 'No Pre-Train w/LG w/AF'). Besides the general promotion compared with the results of random initialization, the edge attribute

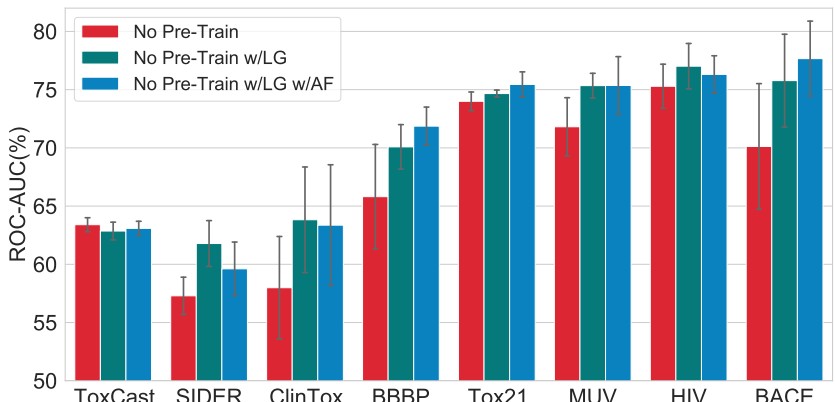

Figure 3: Average test ROC-AUC (%) gain within 'No Pre-Train' from the line graph (w/LG) and edge attribute fusion (w/AF) across all datasets.

Table 2: Average test ROC-AUC (%) of LGCL with different components. Avg. shows the average ROC-AUC over all datasets. A.R. denotes the average rank.

| # | LG | AF | $\mathcal{L}_G$ | $\mathcal{L}_{IntraC}$ | $\mathcal{L}_{InterC}$ | BBBP | Tox21 | ToxCast | SIDER | ClinTox | MUV | HIV | BACE | Avg. | A.R. |
|---|----|----|------|----------|----------|------|-------|---------|-------|---------|-----|-----|------|------|------|
| 1 | ✓ |   | ✓ |   |   | 69.97±2.62 | 75.60±0.50 | 62.70±0.55 | 59.32±1.26 | 67.79±2.52 | 75.70±1.75 | 75.72±0.84 | 82.50±0.73 | 71.16 | 5.4 |
| 2 | ✓ |   | ✓ | ✓ |   | 70.61±1.21 | 75.85±0.49 | 63.83±0.74 | 59.33±0.51 | 75.82±2.35 | 76.37±1.81 | 77.32±0.95 | 80.79±1.41 | 72.49 | 3.6 |
| 3 | ✓ |   | ✓ |   | ✓ | 69.87±1.27 | 74.97±0.52 | 64.01±0.68 | 60.49±1.32 | 76.44±4.32 | 75.52±1.45 | 77.40±1.69 | 84.00±0.66 | 72.84 | 4.0 |
| 4 | ✓ |   | ✓ | ✓ | ✓ | 70.94±0.68 | 75.74±0.63 | 63.50±0.65 | 60.47±1.08 | 76.48±4.82 | 76.12±1.93 | 77.49±0.79 | 84.23±0.87 | 73.12 | 2.9 |
| 5 | ✓ | ✓ | ✓ |   |   | 70.01±0.45 | 75.57±0.56 | 63.59±0.60 | 61.15±0.49 | 74.95±3.27 | 75.97±1.88 | 76.51±1.47 | 83.67±1.49 | 72.68 | 4.1 |
| 6 | ✓ | ✓ | ✓ | ✓ | ✓ | 70.99±1.05 | 76.95±0.43 | 64.71±0.72 | 63.37±0.56 | 77.59±1.54 | 77.70±3.00 | 78.69±1.10 | 84.68±0.73 | 74.33 | 1.0 |

fusion also brings five out of eight better results in contrast to the solo line graph. Furthermore, following the setting of transfer learning, the performance differences between the first and fifth rows as well as the fourth and sixth rows in Table 2 also validate the effectiveness of edge attribute fusion. Thus, we may conclude that edge attribute fusion can alleviate information inconsistency and enhance molecular graph representation learning.

**The effect of intra-local contrastive loss.** The test results under the supervision of the proposed losses are shown in Table 2. To achieve a comprehensive comparison, we first give a baseline only pre-trained with the NT-Xent loss (see the first row). The effectiveness of the proposed intra-local contrastive loss is confirmed by the performance differences between the second and first rows as well as the fourth and third rows, in which the only experimental setup difference is the $\mathcal{L}_{IntraC}$. Specifically, at least six out of eight better results are obtained via deploying this contrastive loss, which informs us of its effectiveness in over-smoothing addressing.

**The effect of inter-local contrastive loss.** Analogously, when comparing the results of the first and third rows as well as the second and fourth rows in Table 2, we can observe that six and five datasets achieve performance exaltation, respectively. This metric promotion indicates the effectiveness of the inter-local contrastive loss in crucial structure identification. Finally, despite several failures within these ablation studies, the last row that simultaneously adopts all proposed components performs best; thus, the proposed parts of LGCL are complementary to each other in molecular semantic exploration regardless of the intricate domain knowledge.

## 6 CONCLUSIONS

In this work, we try to figure out a decent view for molecular graph contrastive learning that can maintain the integrity of molecular semantic information and is friendly to researchers without profound domain knowledge. Inspired by the line graph, we propose a method, called LGCL, to tackle our expectations. Due to the different pace of message passing in the original graph and the corresponding line graph, we further present three crucial components to address the concomitant issues and enhance molecular graph representation learning. Under the setting of transfer learning, we empirically present the superior performance of LGCL over SOTA works.

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

**Supplementary Materials for**

**Look in The Mirror: Molecular Graph Contrastive Learning with Line Graph**

## A  DETAILS OF MOLECULAR DATASETS

As discussed in Section 4.1, a vertex with $e$ edges in $G$ will produce $e \times (e-1)/2$ edges in $L(G)$, which could lead to severe runtime complexity when the original graphs are dense. Therefore, our method only suits sparse graphs. As for molecules adopted in this work, we can see from Table A.1 that the computation of the line graph encoder is 1.5 times that of the original graph encoder according to the average degrees in the transformed line graphs. However, as for the realistic time required for model pre-training, detailed comparisons are shown in Appendix C.2.

**Input graph representation.**  For simplicity, we use a minimal set of node and bond features that unambiguously describe the two-dimensional structure of molecules. We use RDKit Landrum (2013) to obtain these features.

- Node features:
  - Atom number: [1, 118]
  - Chirality tag: {unspecified, tetrahedral cw, tetrahedral ccw, other}
- Edge features:
  - Bond type: {single, double, triple, aromatic}
  - Bond direction: {–, endupright, enddownright}

Table A.1: Datasets statistics summary.

| Dataset | #Tasks | #Graphs | Avg.Node | Avg.Degree | LG Avg.Degree |
|---------|--------|---------|----------|------------|---------------|
| ZINC15  |        | 2,000,000 | 26.63 | 57.72 | 80.98 |
| BBBP    | 1      | 2,039   | 24.06    | 51.90      | 75.10 |
| Tox21   | 12     | 7,831   | 18.57    | 38.58      | 53.03 |
| ToxCast | 617    | 8,576   | 18.78    | 38.52      | 52.95 |
| SIDER   | 27     | 1,427   | 33.64    | 70.71      | 99.97 |
| ClinTox | 2      | 1,477   | 26.15    | 55.76      | 79.76 |
| MUV     | 17     | 93,087  | 24.23    | 52.55      | 73.06 |
| HIV     | 1      | 41,127  | 25.51    | 54.93      | 78.42 |
| BACE    | 1      | 1,513   | 34.08    | 73.71      | 105.78 |

**Downstream task datasets.**  8 binary graph classification datasets from MoleculeNet Wu et al. (2018) are used to evaluate model performance.

- BBBP Martins et al. (2012). Blood-brain barrier penetration (membrane permeability), involves records of whether a compound carries the permeability property of penetrating the blood-brain barrier.
- Tox21 Tox (2014). Toxicity data on 12 biological targets, which has been used in the 2014 Tox21 Data Challenge and includes nuclear receptors and stress response pathways.
- ToxCast Richard et al. (2016). Toxicology measurements based on over 600 in vitro high-throughput screenings.
- SIDER Kuhn et al. (2016). Database of marketed drugs and adverse drug reactions (ADR), grouped into 27 system organ classes and also known as the Side Effect Resource.
- ClinTox Novick et al. (2013); Gayvert et al. (2016). Qualitative data classifying drugs approved by the FDA and those that have failed clinical trials for toxicity reasons.
- MUV Gardiner et al. (2011). Subset of PubChem BioAssay by applying a refined nearest neighbor analysis, designed for validation of virtual screening techniques.

- HIV HIV. Experimentally measured abilities to inhibit HIV replication.
- BACE Subramanian et al. (2016). Qualitative binding results for a set of inhibitors of human $\beta$-secretase 1.

**Dataset splitting.** For molecular prediction tasks, following Ramsundar et al. (2019), we cluster molecules by scaffold (molecular graph substructure) Bemis & Murcko (1996), and recombine the clusters by placing the most common scaffolds in the training set, producing validation and test sets that contain structurally different molecules. Prior work has shown that this scaffold split provides a more realistic estimate of model performance in prospective evaluation compared to random split Chen et al. (2012); Sheridan (2013). The split for train/validation/test sets is 80%:10%:10%.

# B    MOLECULAR PROPERTY PREDICTION VIA UNSUPERVISED LEARNING AND SEMI-SUPERVISED LEARNING

As discussed above, LGCL only suits sparse graphs to avoid too bad runtime complexity. As shown in Table A.2, the capacities of all social network datasets and dense bioinformatics dataset (i.e., DD) heavily increase after the line graph transformation, which leads to unaffordable computation consumption in graph representation learning. Therefore, we only employ sparse bioinformatics datasets for unsupervised and semi-supervised learning. Experiment details are elaborated below.

Table A.2: Summary statistics of ubiquitous benchmarks from TUDataset.

| Dataset | #Graphs | #Classes | Avg.Nodes | Avg.Degree | LG Avg.Degree |
|---|---|---|---|---|---|
| Social Networks | | | | | |
| COLLAB | 5,000 | 3 | 74.49 | 4914.43 | 786967.36 |
| REDDIT-BINARY | 2,000 | 2 | 429.63 | 995.50 | 184826.67 |
| REDDIT-MULTI-5K | 4,999 | 5 | 508.52 | 1189.74 | 81066.29 |
| IMDB-BINARY | 1,000 | 2 | 19.77 | 193.06 | 2782.11 |
| IMDB-MULTI | 1,500 | 3 | 13.00 | 131.87 | 2037.64 |
| GITHUB | 12,725 | 2 | 113.79 | 469.27 | 19574.33 |
| Bioinformatics | | | | | |
| MUTAG | 188 | 2 | 17.93 | 39.58 | 57.74 |
| NCI1 | 4,110 | 2 | 29.87 | 64.60 | 93.21 |
| PROTEINS | 1,113 | 2 | 39.06 | 145.63 | 448.99 |
| DD | 1,178 | 2 | 284.32 | 1431.31 | 6581.59 |

## B.1    UNSUPERVISED LEARNING

**Datasets.** Three sparse bioinformatics datasets are adopted from TUDataset Morris et al. (2020) for unsupervised learning, including NCI1 and MUTAG, and PROTEINS. Table A.2 summarizes the characteristics of the three employed datasets.

- NCI1 is a dataset made publicly available by the National Cancer Institute (NCI) and is a subset of balanced datasets containing chemical compounds screened for their ability to suppress or inhibit the growth of a panel of human tumor cell lines; this dataset possesses 37 discrete labels.
- MUTAG has seven kinds of graphs that are derived from 188 mutagenic aromatic and heteroaromatic nitro compounds.
- PROTEINS is a dataset where the nodes are secondary structure elements (SSEs), and there is an edge between two nodes if they are neighbors in the given amino acid sequence or in 3D space. The dataset has 3 discrete labels, representing helixes, sheets or turns.

**Configuration.** To keep in line with GraphCL You et al. (2020), the same GNN architectures are employed with their original hyper-parameters under individual experiment settings. Specifically, GIN Xu et al. (2019) with 3 layers is set up in unsupervised representation learning. The encoder

hidden dimensions are fixed for all layers to keep in line with GraphCL under individual experiment setting. Models are trained 20 epochs and tested every 10 epochs. Hidden dimension is 32, and batch size is $\in \{32, 128\}$. An Adam optimizer Kingma & Ba (2015) is employed to minimize the contrastive lose and learning rate is $\in \{0.01, 0.001, 0.0001\}$.

**Learning protocol.** Following the learning setting in SOTA works, the corresponding learning protocols are adopted for a fair comparison. In unsupervised representation learning Sun et al. (2020), all data is used for model pre-training and the learned graph embeddings are then fed into a non-linear SVM classifier to perform classification. Experiments are performed for 5 times each of which corresponds to a 10-fold evaluation as Sun et al. (2020), with mean and standard deviation of accuracies (%) reported.

**Compared methods.** We adopt nine baselines that are composed of three categories. The published hyper-parameters of these methods are adopted. The first set is three SOTA kernel-based methods that include GL Shervashidze et al. (2009), WL Shervashidze et al. (2011) and DGK Yanardag & Vishwanathan (2015). The second set is four heuristic self-supervised methods, including node2vec Grover & Leskovec (2016), sub2vec Adhikari et al. (2018), graph2vec Annamalai Narayanan & Jaiswal (2017), and InfoGraph Sun et al. (2020). The final compared methods are GraphCL You et al. (2020), JOAO(v2) You et al. (2021), AD-GCL Suresh et al. (2021), AutoGCL Yin et al. (2022) and RGCL Li et al. (2022b).

**Results.** The results of LGCL along with SOTA competitors on three benchmarks are shown in Table A.3. To summarize, the proposed graph contrastive learning framework with the line graph, LGCL, obtains superior performance compared with the previous works. In particular, except for NCI1, LGCL achieves the best performance on two out of three benchmarks. Thus, we can conclude that LGCL captures the molecular semantic information well in the setting of unsupervised learning.

Table A.3: Average accuracies (%) $\pm$ Std. of compared methods via unsupervised learning. **Bold** indicates the best performance over all methods. *Italic* marks the second best performance.

|  | NCI1 | PROTEINS | MUTAG |
|---|---|---|---|
| GL | 62.33±0.3 | 71.75±0.6 | 81.66±2.11 |
| WL | 80.01±0.50 | 72.92±0.56 | 80.72±3.00 |
| DGK | *80.31±0.46* | 73.30±0.82 | 87.44±2.72 |
| node2vec | 54.89±1.61 | 57.49±3.57 | 72.63±10.2 |
| sub2vec | 52.84±1.47 | 53.03±5.55 | 61.05±15.8 |
| graph2vec | 73.22±1.81 | 73.30±2.05 | 83.15±9.25 |
| InfoGraph | 76.20±1.06 | 74.44±0.31 | *89.01±1.13* |
| GraphCL | 77.87±0.41 | 74.39±0.45 | 86.80±1.34 |
| JOAO | 78.07±0.47 | 74.55±0.41 | 87.35±1.02 |
| JOAOv2 | 78.36±0.53 | 74.07±1.10 | 87.67±0.79 |
| AD-GCL | 75.86±0.62 | 75.04±0.48 | 88.62±1.27 |
| AutoGCL | **82.00±0.29** | *75.80±0.36* | 88.64±1.08 |
| RGCL | 78.14±1.08 | 75.03±0.43 | 87.66±1.01 |
| LGCL | 79.76±0.46 | **76.47±0.39** | **90.32±1.14** |

## B.2 SEMI-SUPERVISED LEARNING

**Datasets.** Two sparse bioinformatics datasets are adopted from TUDataset Morris et al. (2020) for semi-supervised learning, including NCI1 and PROTEINS.

**Configuration.** ResGCN with 128 hidden units and 5 layers is set up in semi-supervised learning. For all datasets we perform experiments with 10% label rate for 5 times, each of which corresponds to a 10-fold evaluation as You et al. (2020), with mean and standard deviation of accuracies (%) reported. For pre-training, learning rate is tuned in $\{0.01, 0.001, 0.0001\}$ and epoch number in $\{20, 40, 60, 80, 100\}$ where grid search is performed. For fine-tuning, we following the default

setting in You et al. (2020), that is, learning rate is 0.001, hidden dimension is 128, bath size is 128, and the pre-trained models are trained 100 epochs.

**Learning protocols.** Following the learning setting in SOTA works, the corresponding learning protocols are adopted for a fair comparison. In semi-supervised learning You et al. (2020), there exist two learning settings. For datasets with a public training/validation/test split, pre-training is performed only on training dataset, finetuning is conducted with 10% of the training data, and final evaluation results are from the validation/test sets. For datasets without such splits, all samples are employed for pre-training while finetuning and evaluation are performed over 10 folds.

**Compared methods.** Under the setting of semi-supervised learning, eight baselines are adopted, including (1) the naive GCN without pre-training You et al. (2020), which is directly trained with 10% labeled data from random initialization; (2) GAE Kipf & Welling (2016), a predictive method by edge-based reconstruction in the pre-training phase; (3) Infomax Velickovic et al. (2019), a node embedding method with global-local representation consistency; (4) ContextPred Hu et al. (2020), a method via sub-structure information preserving; (5) GraphCL You et al. (2020), the first graph contrastive learning method with data augmentations. (6) JOAO(v2) You et al. (2021), an optimization framework to automatically select data augmentations; (7) AD-GCL Suresh et al. (2021), a framework aim to exclude redundant information during the training by optimizing adversarial graph augmentation strategies; (8) AutoGCL Yin et al. (2022), a model with learnable graph view generators orchestrated by an auto augmentation strategy.

**Results.** The results of LGCL along with SOTA competitors on the two benchmarks are shown in Table A.4, in which LGCL suppresses the SOTA view generation works on the two employed datasets. Thus, we can conclude that LGCL captures the molecular semantic information well in the setting of semi-supervised learning.

Table A.4: Average accuracies (%) $\pm$ Std. of compared methods via semi-supervised representation learning with 10% labels. **Bold** indicates the best performance over all methods. *Italic* marks the second best performance.

|              | NCI1              | PROTEINS          |
| ------------ | ----------------- | ----------------- |
| No Pre-Train | 73.72±0.24        | 70.40±1.51        |
| GAE          | 74.36±0.24        | 70.51±0.17        |
| Infomax      | 74.86±0.26        | 72.27±0.40        |
| ContextPred  | 73.00±0.30        | 70.23±0.63        |
| GraphCL      | 74.63±0.25        | 74.17±0.34        |
| JOAO         | 74.48±0.27        | 72.13±0.92        |
| JOAOv2       | 74.86±0.39        | 73.31±0.48        |
| AD-GCL       | *75.18±0.31*      | 73.96±0.47        |
| AutoGCL      | 73.33±2.86        | *74.57±3.29*      |
| LGCL         | **75.82±0.28**    | **74.87±0.39**    |

## C  FURTHER ABLATION STUDY

### C.1  HYPER-PARAMETER SENSITIVITY

In the design of LGCL, besides the general hyper-parameters (i.e., learning rate, batch size, dropout ratio, etc.), we introduce two hyper-parameters, $\alpha$ and $\beta$, for loss balance in pre-training stage. To clearly show the essential effectiveness of the two losses rather than the two hyper-parameters, we fix $\alpha$ and $\beta$ to 1 in the main text. The other hyper-parameters in pre-training phase are also fixed and consistent with Hu et al. (2020).

To further inspect the hyper-parameter sensitivity of LGCL, we tuned the candidates of $\alpha$ and $\beta$ in the range of [0.01, 0.1, 1, 10, 100], respectively. In the tunning of $\alpha$, we fix the $\beta$ to 1 and vice versa. In the fine-tuning stage, the learning is fixed to 0.001; the batch size is fixed to 32; the dropout ratio is fixed to 0.5. The node representations for graph pooling are adopted from the last layer. All

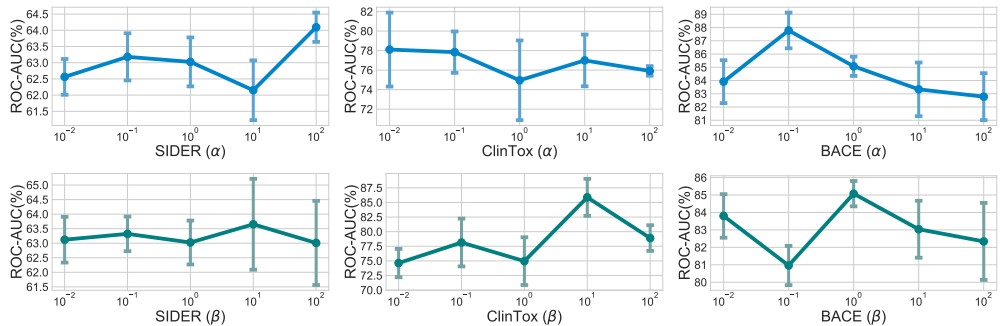

Figure A.1: **Sensitivity *w.r.t.* hyper-parameter $\alpha$ and $\beta$.**

experiments on each dataset are performed for ten runs with different seeds, and the results are the averaged ROC-AUC scores (%) ± standard deviations.

The average ROC-AUC scores of downstream tasks are shown in Figure A.1. As can be seen, different downstream tasks prefer different loss controls. Specifically, ClinTox and BACE prefer small $\alpha$ but large $\beta$, while SIDER would like large $\alpha$ and is insensitive to $\beta$, which suggests that ClinTox and BACE suffer more from the over-smoothing. It is worth noting that more superior results can be obtained from tuning these two hyper-parameters, such as the 85.88% of ClinTox with $\alpha = 1$ and $\beta = 10$ compared to 77.59% in the main text with $\alpha = 1$ and $\beta = 1$, which implies the huge potential of LGCL with the tuning of the two hyper-parameters.

## C.2 EFFICIENCY

In this paper, to address the issues of molecular semantics alteration and generalization capability in molecular contrastive learning, we introduce the line graph and further propose edge attribute fusion, intra-local contrastive loss, and inter-local contrastive loss to enhance contrastive learning. In Section 5, we have shown the superior performance of LGCL. Here, we further present the efficiency of LGCL by comparing the pre-training time on 2 million molecular graphs of ZINC15 with the baselines. In addition, the actual runtime of LGCL through introducing each item is also given, including LGCL with only line graph (i.e., LGCL w/o AF $\mathcal{L}_{IntraC}$ $\mathcal{L}_{InterC}$); LGCL with line graph and edge attribute fusion (i.e., LGCL w/o $\mathcal{L}_{IntraC}$ $\mathcal{L}_{InterC}$); LGCL with line graph, edge attribute fusion and intra-local contrastive loss (i.e., LGCL w/o $\mathcal{L}_{InterC}$). The setting of pre-training time comparison is consistent with the main text Hu et al. (2020). All baselines and LGCL are rerun in our platform (Tesla V100 GPU and Intel(R) Xeon(R) Silver 4214 CPU). Note that, we set the *num_workers* in *dataloader* to *default* for fair comparison.

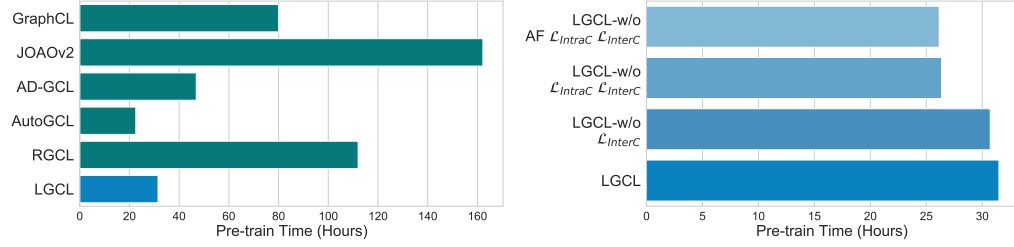

(a) Comparison between baselines and LGCL.     (b) Comparison among parts of LGCL.

Figure A.2: **Pre-training time comparison.** The time required by LGCL is much less than the time needed by baselines except AutoGCL.

Figure A.2 shows the pre-training time required for 2 million molecular graphs from ZINC15 pre-training with 100 epochs. As shown in Figure A.2a, because the contrastive views of LGCL are static, the time required by LGCL is much less than the time needed by baselines except AutoGCL, which reveals that LGCL has not only superior performance but also excellent efficiency. In partic-

ular, LP-Info requires almost 16 hours for one epoch pre-training, that is about 1,600 hours for 100 epochs, thus we do not report its pre-training time in Figure A.2a.

Furthermore, as shown in Figure A.2b, comparing to the model only with line graph (i.e., LGCL w/o AF $\mathcal{L}_{IntraC}$ $\mathcal{L}_{InterC}$), the additional time from the introduction of edge attribute fusion is nearly negligible. The biggest time consumption gap comes from the proposal of two local contrastive losses, but they still only occupy 16.26% of the total time, which is a small price in contrast to the accompanying performance boosting.

## C.3 VISUALIZATION OF OVER-SMOOTHING ALLEVIATING

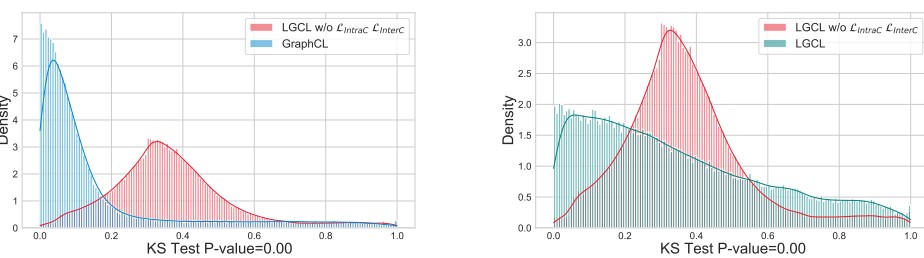

(a) Node similarity distribution of GraphCL and LGCL w/o $\mathcal{L}_{IntraC}$ & $\mathcal{L}_{InterC}$.

(b) Node similarity distribution of GraphCL and LGCL.

Figure A.3: **Visualization of intra-local contrastive loss in over-smoothing alleviating.** We examine the two groups of distributions by a Kolmogorov–Smirnov test (KS test), where the KS test p-values show that the two groups of distributions are distinct (i.e., the p-values are less than 0.01).

In the ablation study, we empirically inspect the effect of intra-local contrastive loss in over-smoothing alleviating. Here, a visualization case study is presented to further directly show how the intra-local contrastive works. Specifically, we calculate the similarities among nodes within the same graph from the pre-training dataset via the model after 100-epoch pre-training. As shown in Figure A.3, we give the node similarity distribution of GraphCL, LGCL, and LGCL without the two local contrastive losses. First, based on the distribution shown in Figure A.3a, we can see that the introduction of line graph leads to a right shift of node similarity distribution, that is, over-smoothing. Then, the left shift of node similarity distribution in Figure A.3b suggests that the proposed loss could alleviate the over-smoothing.

## C.4 VISUALIZATION OF INFORMATION INCONSISTENCY ALLEVIATING

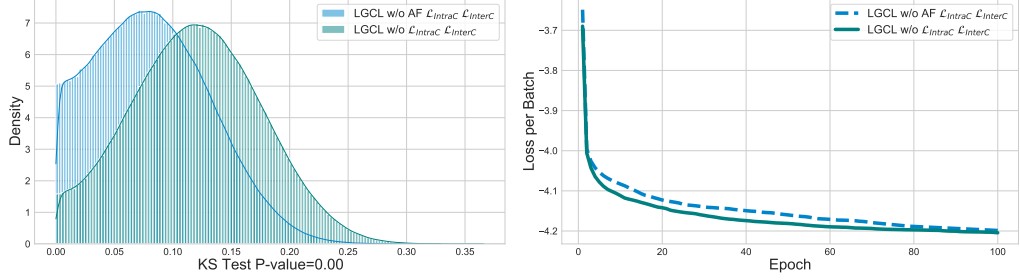

(a) Positive view similarity distributions of LGCL with and without edge attribute fusion.

(b) Pre-traning loss curves of LGCL with and without edge attribute fusion.

Figure A.4: **Visualization of edge attribute fusion in information inconsistency alleviating.** We examine the two view similarity distributions by a Kolmogorov–Smirnov test (KS test), where the KS test p-values show that the two groups of distributions are distinct (i.e., P-value $\leq$ 0.01).

In the ablation study, we empirically validate the effectiveness of edge attribute fusion in information inconsistency alleviating. Here, visualization case studies are further presented to show how edge attribute fusion works. First, we calculate the view similarity of positive pairs from the pre-training dataset via the model after 100-epoch pre-training. As shown in Figure A.4a, we present the view similarity distribution of LGCL with and without the edge attribute fusion [2]. We can see that the introduction of edge attribute fusion significantly increases the view similarity between positive pairs. Second, we inspect the convergence of NT-Xent loss of LGCL with and without edge attribute fusion. As shown in Figure A.4b, LGCL equipped with edge attribute fusion converges faster and has a lower loss, which suggests the higher alignment of the view embeddings. In summary, edge attribute fusion can address the information inconsistency accompanying the line graph.

# D    OTHER BASELINES

Here, we further compare LGCL with other self-supervised molecular representation learning:

- GROVER Rong et al. (2020) predicts the contextual properties based on atom embeddings to encode contextual information into node embeddings.
- MGSSL Zhang et al. (2021) aims to capture the rich information in graph motifs.
- 3D-Infomax Stärk et al. (2022) aims to reason about the geometry of molecules given only their 2D molecular graphs by pre-training model with existing 3D molecular datasets.

The results are shown in Table A.5. As can be seen, LGCL suppresses GROVER on all datasets, MGSSL on six out of eight datasets, and 3D-Infomax on six out of seven datasets. Moreover, LGCL also achieves the highest average results among the these baselines.

Table A.5: Average test ROC-AUC (%) $\pm$ Std. over different 10 runs of LGCL along with all baselines on eight downstream molecular property prediction benchmarks. The results of baselines are derived from the published works. **Bold** indicates the best performance among all baselines. Avg. shows the average ROC-AUC over all datasets. A.R. denotes the average rank. - indicates the data missing in the such works.

| Dataset | BBBP | Tox21 | ToxCast | SIDER | ClinTox | MUV | HIV | BACE | Avg. |
|---|---|---|---|---|---|---|---|---|---|
| GROVER | 68.0±1.5 | 76.3±0.6 | 63.4±0.6 | 60.7±0.5 | 76.9±1.9 | 75.8±1.7 | 77.8±1.4 | 79.5±0.8 | 72.3 |
| MGSSL(DFS) | 70.5±1.1 | 76.4±0.4 | 63.8±0.3 | 60.5±0.7 | 79.7±2.2 | 78.1±1.8 | **79.5±1.1** | 79.7±0.8 | 73.5 |
| MGSSL(BFS) | 69.7±0.9 | 76.5±0.3 | 64.1±0.7 | 61.8±0.8 | 80.7±2.1 | **78.7±1.5** | 78.8±1.2 | 79.1±0.9 | 73.7 |
| 3D-Infomax | 69.1±1.07 | 74.46±0.74 | 64.41±0.88 | 53.37±3.34 | 59.43±3.21 | - | 76.08±1.33 | 79.42±1.94 | - |
| 3D-Infomax+ | 68.64±2.19 | 73.73±0.69 | 63.95±0.38 | 58.43±1.28 | **83.59±3.64** | - | 75.38±0.95 | 79.28±3.61 | - |
| LGCL | **70.99±1.05** | **76.95±0.43** | **64.71±0.72** | **63.37±0.56** | 77.59±1.54 | 77.70±3.00 | 78.69±1.10 | **84.68±0.73** | **74.33** |

# E    THEORETICAL UNDERSTANDING OF LGCL

Besides the superior performance of LGCL shown in the main text for molecular property prediction, here, we further present a theoretical understanding of how LGCL obtains better performance.

**Definition E.1.** (Graph Quotient Space). Define the equivalence $\cong$ between two graphs $G_1 \cong G_2$ if $G_1, G_2$ cannot be distinguished by the 1-WL test. Define the quotient space $\mathcal{G} = \mathcal{G}/\cong$.

So every element in the quotient space, i.e., $G \in \mathcal{G}$, is a representative graph from a family of graphs that cannot be distinguished by the 1-WL test. Note that our definition also allows attributed graphs.

**Theorem E.2.** *Suppose $\mathbb{G}$ is a countable space and thus $\mathbb{G}'$ is a countable space. Because $\mathbb{G}$ and $\mathbb{G}'$ are countable, $\mathbb{P}_{\mathbb{G}}$ and $\mathbb{P}_{\mathbb{G}'}$ are defined over countable sets and thus discrete distribution. Later we may call a function $z(\cdot)$ can distinguish two graphs $G_1, G_2$ if $z(G_1) \neq z(G_2)$. Moreover, for notational simplicity, we consider the following definition. Suppose the encoder $f$ is implemented by a GNN. The optimal encoder $f^*$ is the best model which GNN can find. Because $f^*$ is as powerful as the 1-WL test. Then, for any two graphs $G_1, G_2 \in \mathbb{G}$, $G_1 \cong G_2$, $f^*(G_1) = f^*(G_2)$. We may define a mapping over $\mathbb{G}'$, also denoted by $f^*$ which simply satisfies $f^*(G') :\triangleq f^*(G)$, where $G \cong G'$, and $G \in \mathcal{G}$ and $G' \in \mathcal{G}'$. Suppose $t(\cdot)$ is the data augmentation function and $L(\cdot)$ is the line graph transformation function. We have*

---

[2]The two models are both not equipped with the two local contrastive losses.

1. $I(L(G); G) \geq I(t(G'); G')$;

2. $I(L(G); Y) \geq I(t(G); Y)$.

The statement 1 in Theorem E.2 indicates that the retained information after line graph transformation from given graphs is more than contrastive views with data augmentation.

The statement 2 in Theorem E.2 suggests that the essential information underlying the line graph for target prediction is more than contrastive views with data augmentation.

*Proof.* Given $G$, $G \Rightarrow L(G)$ is an injective deterministic mapping. Therefore, for any random variable $Q$,

$$I(L(G); Q) = I(G; Q). \tag{13}$$

Of course, we may set $Q = G$. So,

$$I(L(G); G) = I(G; G). \tag{14}$$

Then, we have

$$I(L(G'); G') = I(G'; G')$$
$$\overset{(a)}{\geq} I(t(G'); G'), \tag{15}$$

where $(a)$ is because the data processing inequality Cover (1999). Moreover, because $f^*$ could be as powerful as the 1-WL test and is injective in $\mathcal{G}'$. Meanwhile, as stated in the Whitney graph isomorphism theorem Whitney (1932), the isomorphism of two line graphs is judged to be consistent with the corresponding two original graphs, thus we have

$$I(L(G'); G') = I(f^*(L(G')); f^*(G'))$$
$$= I(f^*(L(G)); f^*(G))$$
$$= I(L(G); G). \tag{16}$$

Here, the second equality is because the transformation of line graph will not change the isomorphic relationship between two graphs $G'$ and $G$, meanwhile $f^*(G') = f^*(G)$. Therefore, we achieve the statement 1:

$$I(L(G); G) \geq I(t(G'); G'). \tag{17}$$

Again, because by definition $f^* = argmax_f I(f(G); G)$, $f^*$ must be injective. Given $G^*$, $G^* \Rightarrow f^*(G^*)$ is an injective deterministic mapping. Of course, we may set $Q = Y$. So,

$$I(f^*(G); Y) = I(G^*; Y). \tag{18}$$

Because $G \Rightarrow L(G)$ is an injective deterministic mapping.

$$I(f^*(G); Y) = I(f^*(L(G)); Y),$$
$$= I(L(G); Y). \tag{19}$$

Further because of the data processing inequality Cover (1999),

$$I(f^*(G); Y) = I(G; Y)$$
$$\geq I(t(G); Y)$$
$$= I(f^*(t(G)); Y). \tag{20}$$

Combining above equations, we have the statement 2:

$$I(L(G); Y) = I(f^*(L(G)); Y)$$
$$= I(f^*(G); Y)$$
$$\geq I(f^*(t(G)); Y)$$
$$= I(t(G); Y), \tag{21}$$

which concludes the proof of the essential information of line graph.

$\square$

