# OpenReview forum: "Look in The Mirror: Molecular Graph Contrastive Learning with Line Graph"
_ICLR.cc/2023/Conference — Submitted to ICLR 2023_

### Official Review · Reviewer_3j8n · 2022-10-21

**Confidence:** 4
**Correctness:** 2
**Technical Novelty And Significance:** 3
**Empirical Novelty And Significance:** 3
**Recommendation:** 6

**Clarity, Quality, Novelty And Reproducibility:**

The paper is clear to read and the proposed idea is novel with respect to the existing baselines. Reproducibility of this work will be high if the authors publish their code given acceptance (along with the protocols for selecting the hyperparameters).

**Strength And Weaknesses:**

Strength 1: The idea of using line graph as an additional view for a molecule is novel for pretraining.

Strength 2: The proposed method shows strong empirical performance.

Weakness 1: Most importantly, the proposed method is combination of many ideas, but the current experiments are insufficient to demonstrate significance of such ideas.

Weakness 1.1: In the ablation studies, simply combining line graph and contrastive learning achieves average performance of 71.16, which is worse than six of the considered baselines.

Weakness 1.2: After combining edge attribute fusion, the proposed method generates property prediction using different neural network architecture compared to the baselines. If this is the case, the authors should consider the same architecture for the baselines too.

Weakness 1.3: The proposed method is based on combination of three loss functions, while most of the baselines rely on one loss function. To measure the significance of these loss functions, the experiments should be constructed more carefully. For example, one could fix the budget for tuning the hyperparameters and try to combine the previously proposed pretraining objectives.

Weakness 2: Table 2 does not have any “overall” statistics like ranking and average, so it is hard to interpret the table and make conclusion.

Weakness 3: I suggest comparing with 3D-infomax [1] and MGSSL [2] as a pretraining baseline. I also suggest comparing with works that simultaneously propose graph neural network (or Transformer) architectures with pretraining objective (GROVER [3], MolR [4]). Comparing with other architecture is important since the work also proposes a new architecture for property prediction.

Weakness 4: I appreciate how the authors showed the candidate hyperparameters for the finetuning experiments. However, there is no detail on which metric is used to choose the hyperparameters. Furthermore, it is not clear whether if the budget used for tuning the pretraining hyperaparameter is the comparable across the baselines.

[1] 3D Infomax improves GNNs for Molecular Property Prediction, ICML 2022
[2] Motif-based Graph Self-Supervised Learning for Molecular Property Prediction, ICML 2022
[3] Self-Supervised Graph Transformer on Large-Scale Molecular Data, NeurIPS 2020
[4] Chemical-Reaction-Aware Molecule Representation Learning, ICLR 2022

**Summary Of The Paper:**

This paper proposes a new framework for molecular property prediction. The proposed framework consists of (1) a new edge attribute fusion-based graph neural network architecture and (2) three pretraining objectives. The newly proposed GNN architecture additionally uses line graph-based view of a molecule as an additional source of information. Furthermore, the contrastive learning objectives consist of (1) contrastive learning between graph-based and line graph-based molecular representations, (2) intra-contrastive learning between edge representations, and (3) inter-contrastive learning between edge representations. After combining all the components, the proposed method outperforms the existing pretraining frameworks for molecular property prediction.

**Summary Of The Review:**

This paper proposes an interesting idea for pretraining of molecular representations. However, I am concerned by the empirical evaluation which is not thorough enough to validate the significance of the proposed work.

---

> ### Author Response · Authors · 2022-11-14
> **Initial Response to Reviewer 3j8n (part 1)**
>
> **Q1:** Most importantly, the proposed method is combination of many ideas, but the current experiments are insufficient to demonstrate significance of such ideas.
>
> **Answer:** First, the issue discussed in this work is a novel problem that we have concluded based on previous research: how to pre-train molecular representation networks without altering their semantic structures and without relying on domain knowledge. In previous studies, researchers have discussed how to maintain semantics. The conclusion is the introduction of domain knowledge, which will lead to the second issue discussed in this work, the poor generalization capability because domain knowledge varies from domains. In this context, we seek to solve this problem in this paper, namely, molecular contrastive learning without destroying the semantic structure and relying on domain knowledge.
>
> Second, to solve this problem, this paper introduces the concept of line graph, which has indeed been applied in many tasks regarding graph representation learning, which is why we choose the line graph, that is, its excellent information representation ability. At the same time, if we just introduce line graphs, we cannot suppress baselines, which is also what we discussed in the method section. The introduction of line graphs brings two additional problems: over-smoothing and inconsistent information. Therefore, based on the introduction of line graphs, we also propose edge attribute fusion and two local losses to enhance the expressiveness of line graphs and improve the effectiveness of the model on molecular contrastive learning. For details, please refer to the ablation study in the main text.
>
> **Q1.1:** In the ablation studies, simply combining line graph and contrastive learning achieves average performance of 71.16, which is worse than six of the considered baselines.
>
> **Answer:** Yes, this is also the answer to Q1. Only introducing the line graph is not enough. It also brings two additional problems, so the expressive power of the line graph needs to be enhanced by additional parts, that is, the edge attribute fusion and our two local contrastive losses. If superior results can be easily obtained via only introducing line graphs, it is believed that some researchers have already published this work.
>
> **Q1.2:** After combining edge attribute fusion, the proposed method generates property prediction using different neural network architecture compared to the baselines. If this is the case, the authors should consider the same architecture for the baselines too.
>
> **Answer:** Actually, the novel framework of LGCL is also one of the contributions of our paper. In addition, the edge attribute fusion and losses we proposed can only be used within the framework of LGCL because the graph and the corresponding line graph have a one-to-one correspondence between nodes and edges, but the baseline methods adopt the data augmentation to change the topology of input molecules, which leads to the loss of such correspondence. Therefore, baselines do not fit the proposed architecture.
>
> **Q1.3:** The proposed method is based on combination of three loss functions, while most of the baselines rely on one loss function. To measure the significance of these loss functions, the experiments should be constructed more carefully. For example, one could fix the budget for tuning the hyperparameters and try to combine the previously proposed pretraining objectives.
>
> **Answer:** As discussed in the answer to Q1, the proposed losses aim to tackle the issue of over-smoothing and enhance molecular representation learning. Moreover, in the ablation study, we further empirically validate the effectiveness of the proposed losses. Detailed sensitivity experiments regarding the $\alpha$ and $\beta$ of the two losses are also conducted in Appendix C.1 of the new revision.
>
> More importantly, all adopted (molecular contrastive learning) baselines except GraphCL have more than one loss, including the D-SLA introduced by Reviewer xTh3 and the MGSSL mentioned in Q3.
>
> **Q2:** Table 2 does not have any “overall” statistics like ranking and average, so it is hard to interpret the table and make conclusion.
>
> **Answer:** Thanks for noting this. We have fixed this in the new revision.

---

> ### Author Response · Authors · 2022-11-14
> **Initial Response to Reviewer 3j8n (part 2)**
>
> **Q3:** I suggest comparing with 3D-infomax [1] and MGSSL [2] as a pretraining baseline. I also suggest comparing with works that simultaneously propose graph neural network (or Transformer) architectures with pretraining objective (GROVER [3], MolR [4]). Comparing with other architecture is important since the work also proposes a new architecture for property prediction.
>
> **Answer:** Thanks for your suggestions. We include 3D-Infomax, GROVER, MGSSL(DFS), and MGSSL(GFS) as the baselines and attach the results below. Considering the space limitation of the main text, the comparison between these baselines and LGCL is presented in Appendix D. As can be seen, LGCL suppresses GROVER on all datasets, MGSSL on 6/8 datasets, and 3D-Infomax on 6/7 datasets. Moreover, LGCL also achieves the highest average results among the new baselines. Note that, because MolR adopts a different dataset split scheme,  the `random` split, while baselines and LGCL employ the `scaffold` split that provides a more realistic estimate of model performance in prospective evaluation compared to random split. Please refer to Appendix E in Hu et al. ICLR 2020[1]. Moreover, the scaffold split provides more difficult prediction tasks and is the mainstream data split method in the current molecular contrastive learning. Therefore, we do not make a direct comparison with MolR.
>
> -| BBBP |  Tox21 | ToxCast | SIDER | ClinTox | MUV | HIV | BACE | Avg.
> :---:|:---:|:---:|:---:|:---:|:---:|:---:|:---:|:---:|:---:
> GROVER | 68.0 | 76.3 | 63.4 | 60.7 | 76.9 | 75.8 | 77.8 | 79.5 | 72.3
> MGSSL(DFS) | 70.5 | 76.4 | 63.8 | 60.5 | 79.7 | 78.1 | **79.5** | 79.7 | 73.5
> MGSSL(BFS) | 69.7 | 76.5 | 64.1 | 61.8 | 80.7 | **78.7** | 78.8 |  79.1 | 73.7
> 3D-Infomax | 69.1 | 74.46 | 64.41 | 53.37 | 59.43 |  - | 76.08 | 79.42 | -
> 3D-Infomax+ | 68.64 | 73.73 | 63.95 | 58.43 | **83.59** | - | 75.38 | 79.28 | -
> LGCL | **70.99** | **76.95** | **64.71** | **63.37** | 77.59 | 77.70 | 78.69 | **84.68** | **74.33**
>
> It is worth noting that, as suggested by Reviewer awP4, superior results can be obtained by the tunning of $\alpha$ and $\beta$ in the pre-training phase. The updated results of LGCL on BACE, SIDER, and ClinTox are shown below. We will also update the main text while finishing the sensitivity experiments. Please refer to the answer to Q1.1 of Review awP4 for more details.
>
>   -| BACE | SIDER | ClinTox
> :---:|:---:|:---:|:---:
> LGCL in Main Text |84.68$\pm$0.73|63.37$\pm$0.56|77.59$\pm$1.54
> LGCL after Sensitivity |87.78$\pm$1.35($\uparrow$**3.1**)|64.09$\pm$0.45($\uparrow$**0.72**)|85.88$\pm$3.16($\uparrow$**8.29**)
>
> [1] W Hu, B Liu, J Gomes, M Zitnik, P Liang, V Pande, and J Leskovec. Strategies for pre-training graph neural networks. International Conference on Learning Representations (ICLR), 2020.
>
> **Q4:** I appreciate how the authors showed the candidate hyperparameters for the finetuning experiments. However, there is no detail on which metric is used to choose the hyperparameters. Furthermore, it is not clear whether if the budget used for tuning the pretraining hyperaparameter is the comparable across the baselines.
>
> **Answer:** Hyper-parameters are selected based on the average ROC-AUC of the validation set, and the new revision is fixed with the relevant descriptions in the main text. For a fair comparison, we adopt the same hyper-parameters as GraphCL during pre-training. At the same time, we also fix the two introduced parameters $\alpha$ and $\beta$ in the pre-training phase to 1 for the convenience of reproducibility.
>
> To show the efficiency. We rerun baselines and report their pre-training time. Detailed time consumptions of baselines and LGCL with the introduction of each proposed component are discussed in the new revision. Please refer to Appendix C.2 for model efficiency. In summary, because the contrastive views of LGCL are static, the time required by LGCL for 2 million molecular graphs from ZINC15 pre-training (100 epochs) is much less than the time needed by baselines except AutoGCL. Here, we give a brief table for training time comparison under the same setting in our experimental environment. Note that we set the `num_workers` in `dataloader` to `default` for a fair comparison.
> In particular, LP-Info[2] requires too much time to finish its pre-training stage; thus, we only run one epoch and report its time consumption.
>
>  2M Zinc15 | GraphCL | JOAOv2 | LP-Info | AD-GCL | AutoGCL | RGCL | LGCL
> :---:|:---:|:---:|:---:|:---:|:---:|:---:|:---:
> Hours (100 epochs) |79.99|162.20|15h50min (1 epoch)|46.88|22.15|111.99| 31.49

---

> > ### Comment · Reviewer_3j8n · 2022-11-17
> > **Thank you for the detailed response!**
> >
> > I appreciate how the authors provided thoughtful replies to my reviews. Just to rephrase my biggest concerns:
> >
> > 1. (Q1.2) By the nature of LGCL, it requires using a new GNN architecture. However, this is an "unfair" comparison in the sense that previous pretraining methods stick to the GIN architecture constructed by Hu et al., 2017. I am not sure how one can fairly evaluate this, so I suggested GROVER, MGSSL, and MolR (works that use new architecture) as new baselines. To my knowledge, GROVER and MGSSL also use the random split for evaluating the model. Is the newly provided table based on GROVER & MGSSL evaluated on the random split? Are they re-evaluated? If otherwise, I would suggest re-evaluating LGCL on the random split and compare with GROVER, MGSSL, and MGSSL.
> >
> > 2. (Q1.3) My other concern was that LGCL relies on a combination of loss functions and the authors should be more careful about comparing with baselines. Given the author's response, I resonate with how "fairly" compared with the baselines is quite hard with limited computing power. Nevertheless, I would suggest the authors include a brief discussion about the limitations of the current evaluation protocols in their manuscript.
> >
> > If my suggestions are incorporated (or I become persuaded by how they are not necessary), I will raise my score.

---

> > > ### Author Response · Authors · 2022-11-17
> > > **Thanks for the follow up**
> > >
> > > Thank you for the follow up and helpful suggestions.
> > >
> > > **Q1.2:** By the nature of LGCL, it requires using a new GNN architecture. However, this is an "unfair" comparison in the sense that previous pretraining methods stick to the GIN architecture constructed by Hu et al., 2017. I am not sure how one can fairly evaluate this, so I suggested GROVER, MGSSL, and MolR (works that use new architecture) as new baselines. To my knowledge, GROVER and MGSSL also use the random split for evaluating the model. Is the newly provided table based on GROVER & MGSSL evaluated on the random split? Are they re-evaluated? If otherwise, I would suggest re-evaluating LGCL on the random split and compare with GROVER, MGSSL, and MGSSL.
> > >
> > > **Answer:** Thanks for your suggestions. Actually, as mentioned in Section 5.2 (**Pre-training details**) of the main text, the GNN architecture (graph encoder)  is consistent with Hu et al. (2020) [1] and is the GIN [2]:
> > > > In the graph encoder setting in Hu et al. (2020), a Graph Isomorphism Network (`GIN`) Xu et al. (2019) with five convolutional layers is adopted for message passing.
> > >
> > > Therefore, LGCL makes a fair comparison with these pre-training methods regarding the GIN architecture.
> > >
> > > The new results of GROVER and MGSSL in Appendix D are derived from the original paper of MGSSL. In particular, as also mentioned in the original paper of MGSSL, the dataset split method of GROVER and MGSSL is `scaffold split` rather than `random split.` Please refer to the Section 4.1 (**Datasets and Dataset Splittings**) of MGSSL [3]:
> > > > To mimic the real-world use case, we split the downstream dataset by ` scaffold-split` [14, 31], which splits the molecules according to their structures.
> > >
> > > Here, we give the results of LGCL with the random split. In particular, considering the time limit, we fix all hyper-parameters. Specifically, in pre-training, $\alpha=1$, $\beta=1$; in fine-tunning, learning rate=0.001, batch size = 32 and dropout ratio = 0.5.
> > >
> > > -| split | BBBP |  BACE | Tox21 | ClinTox | Avg.
> > > :---:|:---:|:---:|:---:|:---:|:---:|:---:
> > > MolR-GCN | random | 89.0 | 88.2 | 81.8 | 91.6 | 87.65
> > > MolR-GAT | random | 88.7 | 86.3 | 83.9 |  90.8 | 87.42
> > > MolR-SAGE | random | 87.9 | 85.9 | 81.1 | 89.0 | 85.79
> > > MolR-TAG | random |  89.5 | 87.5 | 82.0 | 91.3 | 87.57
> > > LGCL | random | **95.4** | **91.7** | **84.5** | **92.3** | **90.96**
> > >
> > > [1] W Hu, B Liu, J Gomes, M Zitnik, P Liang, V Pande, and J Leskovec. Strategies for pre-training graph neural networks. In ICLR, 2020.
> > >
> > > [2] Keyulu Xu, Weihua Hu, Jure Leskovec, and Stefanie Jegelka. How powerful are graph neural networks? In ICLR, 2019.
> > >
> > > [3] Zaixi Zhang, Qi Liu, Hao Wang, Chengqiang Lu, and Chee-Kong Lee. Motif-based graph self-supervised learning for molecular property prediction. Advances in Neural Information Processing Systems, 34:15870–15882, 2021.
> > >
> > >
> > > **Q1.3:** My other concern was that LGCL relies on a combination of loss functions and the authors should be more careful about comparing with baselines. Given the author's response, I resonate with how "fairly" compared with the baselines is quite hard with limited computing power. Nevertheless, I would suggest the authors include a brief discussion about the limitations of the current evaluation protocols in their manuscript.
> > >
> > > **Answer:** Thanks for noting this. As for the limited computing power, we have mentioned in answer to **last Q1.3**, all adopted (molecular contrastive learning) baselines except GraphCL have more than one loss, including the D-SLA introduced by Reviewer xTh3 and the MGSSL. Moreover, we made a direct comparison of model efficiency. Please refer to the answer to **last Q4** and Appendix C.2 in the new revision.
> > >
> > > Note that the current evaluation protocols are consistent with Hu et al. 2020 [1] and ubiquitous among current molecular contrastive learning methods (such as GraphCL [4], AD-GCL [5], and 3D-Infomax [6]). In my view, current evaluation protocols only evaluate the model performance on molecular property classification, more tasks regarding molecules are supposed to be coverd, such as molecuar regression tasks. This could be a future research direction and future application domain of LGCL.
> > >
> > > [4] Yuning You, Tianlong Chen, Yongduo Sui, Ting Chen, Zhangyang Wang, and Yang Shen. Graph contrastive learning with augmentations. Advances in Neural Information Processing Systems, 33:5812–5823, 2020.
> > >
> > > [5] Susheel Suresh, Pan Li, Cong Hao, and Jennifer Neville. Adversarial graph augmentation to improve graph contrastive learning. Advances in Neural Information Processing Systems, 34, 2021.
> > >
> > > [6] Hannes Sta ̈rk, Dominique Beaini, Gabriele Corso, Prudencio Tossou, Christian Dallago, Stephan Gu ̈nnemann, and Pietro Lio. 3d infomax improves gnns for molecular property prediction. In International Conference on Machine Learning. PMLR, 2022.

---

> > > > ### Comment · Reviewer_3j8n · 2022-11-18
> > > > **Thank you for the detailed respones.**
> > > >
> > > > Now I see that I missed some parts of the literature regarding the splitting strategy. I appreciate the newly updated revision. I increased my scores since my concerns are mostly resolved.

---

### Official Review · Reviewer_xTh3 · 2022-10-21

**Confidence:** 4
**Correctness:** 3
**Technical Novelty And Significance:** 4
**Empirical Novelty And Significance:** 3
**Recommendation:** 8

**Clarity, Quality, Novelty And Reproducibility:**

### Clarity
* The tackled problems (i.e., existing contrastive learning methods alter the semantic structures of graphs, and also sometimes require the domain-specific knowledge) and proposed methods, based on the line graph-based contrastive learning with edge attribute fusion and local contrastive losses, are clearly described.
* However, the clarity of the over-smoothing issue and the experimental setup of Figure 3 should be further improved (See Weaknesses for details).

### Quality
* The technical quality of this work is high. The authors define the under-explored challenges of the existing contrastive learning, and tackle them with the novel line graph-based contrastive learning.
* The experimental results show that the proposed LGCL outperforms baselines, which further supports the technical quality of the proposed idea.

### Novelty
* To my knowledge, the main contrastive learning objective based on the similarity learning between original and line graphs is novel.
* In addition to the main objective, the proposed additional components, such as edge attribute fusion scheme and local contrastive losses, are novel.

### Reproducibility
* The authors do not provide the source code that lowers the reproducibility of this paper; however, the authors plan to release the source code after the acceptance. Therefore, the reproducibility will be probably high.

**Strength And Weaknesses:**

### Strengths
* The idea of using line graph transformation for graph contrastive learning, where the positive pair consists of the original and its line graph, is interesting and novel.
* The problem of information inconsistency between original and line graphs is well justified, which is nicely solved with the proposed edge attribute fusion method.
* The proposed method empirically outperforms other competitive baselines.
* This paper is well-structured, and easy to follow.

### Weaknesses
* There is a recent work [1] that points out the limitation of contrastive learning: altering the semantics of graphs during perturbation, and this problem-level idea is the same as the authors suggest. Thus, this relevant work [1] should be discussed, and might be compared.
* The tackled over-smoothing issue incurred by the line graph is unclear. Why the line graph additionally introduces the over-smoothing issue? And how this over-smoothing issue is tackled by the proposed edge-level contrastive losses in Section 4.3 and Section 4.4?
* The experimental setup of Figure 3 is unclear. Does "No Pre-Train w/LG" denote the fine-tuning model that uses representations of both original and line graphs? Then, for the proposed method, is the READOUT function in equation (3) changed to include the line graph representation?

---

[1] Graph Self-supervised Learning with Accurate Discrepancy Learning, NeurIPS 2022.

**Summary Of The Paper:**

**(Motivation:)** This paper points out the limitations of existing graph contrastive learning, which not only alters the semantic of original graphs during perturbation, but also leverages the domain-specific knowledge that might not be generalizable across different graph datasets.

**(Method:)** To tackle those two problems, the authors propose to use the line graph of the original graph, wherein the edge of the original graph is transformed to the node of the line graph. In particular, the authors maximize the representation similarity of original and its line graph, while minimizing other negative original-line graph pairs, under the existing contrastive learning objective. Also, the authors propose to minimize the inconsistency between node/edge features of the original graph and edge/node features of the line graph, by interchanging their information. Furthermore, the authors propose two additional contrastive losses, which capture edge-level similarities within and between graphs.

**(Experiment:)** The authors validate the proposed LGCL on the graph classification tasks, showing the propose model outperforms relevant baselines.

**Summary Of The Review:**

While there are some revision points: discussing more work and improving clarify of some sections, I believe they require minor revisions. Thus, given the high quality and novelty of this work with strong empirical results, I recommend the acceptance.

---

> ### Author Response · Authors · 2022-11-14
> **Initial Response to Reviewer xTh3**
>
> Thank you very much for the approval of our work and the research philosophy that simplicity is good. We sincerely appreciate your comments, and would be happy to address your concerns below.
>
> **Q1:** There is a recent work [1] that points out the limitation of contrastive learning: altering the semantics of graphs during perturbation, and this problem-level idea is the same as the authors suggest. Thus, this relevant work [1] should be discussed, and might be compared.
>
> **Answer:** Thanks for your kind suggestion. D-SLA [1] addresses the issue of semantic alteration by distinguishing whether the input view is augmented or not. It cleverly converts the problem of semantic alteration into a judgment problem but still introduces the data augmentation method. We attach the comparison results below and in the main text. Please refer to the new revision. It can be seen that LGCL suppresses D-SLA on 6/8 datasets, and the average result is also higher.
>
> -| BBBP |  Tox21 | ToxCast | SIDER | ClinTox | MUV | HIV | BACE | Avg.
> :---:|:---:|:---:|:---:|:---:|:---:|:---:|:---:|:---:|:---:
> D-SLA| **72.6**|76.81| 64.24|60.22|**80.17**|76.64|78.59|83.81|74.14
> LGCL| 70.99 | **76.95** | **64.71** | **63.37** | 77.59 | **77.70** | **78.69** | **84.68**|**74.36**
>
>
> **Q2:** The tackled over-smoothing issue incurred by the line graph is unclear. Why the line graph additionally introduces the over-smoothing issue? And how this over-smoothing issue is tackled by the proposed edge-level contrastive losses in Section 4.3 and Section 4.4?
>
> **Answer:**  Thanks for noting this. Over-smoothing is an inherent issue in deep graph networks. Moreover, due to the exponentially increasing message-passing frequency in line graphs, information spreads more rapidly between nodes, which could worsen the over-smoothing issue. As concerned, in the ablation study, we empirically inspect the effect of intra-local contrastive loss in over-smoothing alleviating. Please refer to the "The effect of intra-local contrastive loss" in the main text. Furthermore, a visualization case study is presented to directly show how the proposed losses work. Specifically, we calculate the similarities among nodes within the same graph from the pre-training dataset via the model after 100-epoch pre-training. Please refer to **Appendix C.3** for details. In summary, the introduction of line graph leads to a right shift of node similarity distribution, that is, over-smoothing. Then, a left shift on node similarity distribution achieved with the introduction of the proposed local losses suggests the effectiveness in over-smoothing alleviating.
>
> **Q3:** The experimental setup of Figure 3 is unclear. Does "No Pre-Train w/LG" denote the fine-tuning model that uses representations of both original and line graphs? Then, for the proposed method, is the READOUT function in equation (3) changed to include the line graph representation?
>
> **Answer:** Sorry for the unclear experimental setup. "No Pre-Train w/LG" denotes the LGCL that is not pre-trained with ZINC15 and is not equipped with the proposed edge attribute fusion and two local contrastive losses. And yes, "No Pre-Train w/LG" uses representations of both original and line graphs, and READOUT in Eq. (3) for LGCL is a concatenation of graph representation and line graph representation.
>
> Again, we appreciate Reviewer xTh3 for the approval of our work and the research philosophy. If there are further concerns, please do let us know. We would be happy to address them.

---

> > ### Comment · Reviewer_xTh3 · 2022-11-26
> > **Thank you for your response**
> >
> > I sincerely thank the authors for providing the response. The authors clarify all the concerns/comments, and, after reading other reviews, I still would like to recommend the acceptance of this work, which innovatively considers the line graph for graph self-supervised learning. Nice work.

---

### Official Review · Reviewer_KhWt · 2022-10-25

**Confidence:** 4
**Correctness:** 3
**Technical Novelty And Significance:** 2
**Empirical Novelty And Significance:** 2
**Recommendation:** 3

**Clarity, Quality, Novelty And Reproducibility:**

## Clarity, Quality, Novelty and Reproducibility

### Clarity
The paper is clearly written and structured and easy to follow.


### Quality

A) The work is not well embedded into related works. It only looks into very closely related works on graph pre-training but not even a bit ahead to other methods for molecular property prediction or to contrastive learning. The authors should embed their work better into related works, and, e.g., check how simple methods and representations usually perform on Tox21[1] or the MoleculeNet benchmarks.

B) The experimental part is limited and all methods operate at performances that are far below usual approaches to molecular property prediction. The authors only perform experiments for molecular property prediction on the so-called MoleculeNet benchmark. Since the method should generally work for graph pre-training, the authors should also perform experiments on graph that arise in a different domain (e.g. documents or social networks). Furthermore, the usual encoding or representation of molecular graphs are extended-connectivity fingerprints [2] that just encode subgraphs of different size. Using this simple and efficient representation, one can easily outperform the presented method using linear probing (see my table below). Thus, one can save all the computational costs that is used for pre-training the suggested encoder.
The authors should include baselines into their table (such as linear probing on frequently used molecular descriptors), reference usual performances on these datasets (e.g. in MoleculeNet), and suggest a pre-training method which at least matches the performance of molecular descriptors.

|          type              |   method                                 | split        | BACE           | BBBP           | ClinTox        | HIV            | MUV            | SIDER          | Tox21          | ToxCast        |
|----------------|--------------------------------|----------|------------|------------|------------|------------|------------|------------|------------|------------|
| Linear probing | ECFP encoder-4096 (reviewer)                 | random   | 91.52±0.0  | 91.25±0.0  | 71.14±0.0  | 80.36±0.0  | 77.05±0.0  | 64.61±0.0  | 76.87±0.0  | 69.43±0.0  |

C) Lack of error bars, confidence intervals and statistical test for many performance metrics.
The performance metrics in Figure 3 and Table 2 are provided withouth error bars or confidence intervals, such that differences could just arise by chance. The authors should perform repeated training runs, or perform cross-validation to obtain error bars.

D) Unclear balancing strategies for the loss function and missing hyperparameter selection procedure.
The authors use a combination of three loss functions to pre-train their GNN. However, such combination is usually hard to balance since the losses can be at completely different scales or move to different scales during learning -- a major problem, e.g., in deep reinforcment learning. Suprisingly, in this work the balancing parameters $\alpha$ and $\beta$ could just be set to 1. Generally, it is unclear how the many hyperparameters that come with this architecture (scale parameters, learning rate schedules, GNN architecture, regularization, ...) have been selected. The authors should describe their hyperparameter selection strategy and their metric how to select good hyperparameters.


### Novelty
The proposed strategy to pre-train graph neural networks appears novel; differences and similarities to other pre-training strategies, e.g. [3], should be described clearer.


### Reproducibilty
The code has not been provided, such that this work is hardly reproducible.



### References
[1] Papers with Code, Tox21 (Drug Discovery), https://paperswithcode.com/sota/drug-discovery-on-tox21
[2] Rogers, D., & Hahn, M. (2010). Extended-connectivity fingerprints. Journal of chemical information and modeling, 50(5), 742-754.
[3] Fang, R., Wen, L., Kang, Z., & Liu, J. (2022). Structure-Preserving Graph Representation Learning. arXiv preprint arXiv:2209.00793.

**Strength And Weaknesses:**

Strengths:
- The work approaches a relevant problem, how to pre-train molecular encoders
- Clearly written and structured paper

Weaknesses:
- The lacks embedding into related works outside of this small community of graph pre-training
- The claimed improvement is hardly relevant since even the most simplistic molecular descriptors (Morgan, ECFP) produce better representations
- Only a single experiment has been performed
- There are technical errors, such as the lack of error bars and confidence intervals, and unclear model and hyperparameter selection



**Summary Of The Paper:**

The paper proposes a pre-training strategy for graph neural networks based on line graphs. The method is then benchmarks on several molecular property prediction tasks.

**Summary Of The Review:**

The proposed pre-training strategy is reasonably novel, altough not well embedded into related work. The results and conclusions are hardly relevant since the predictive quality is even below classical molecular descriptors. There are several technical errors in the experimental part.

---

> ### Author Response · Authors · 2022-11-14
> **Initial Response to Reviewer KhWt (part 1)**
>
> **Q1:** The work is not well embedded into related works. It only looks into very closely related works on graph pre-training but not even a bit ahead to other methods for molecular property prediction or to contrastive learning. The authors should embed their work better into related works, and, e.g., check how simple methods and representations usually perform on Tox21[1] or the MoleculeNet benchmarks.
>
> **Answer:** Thanks for noting this. Molecular contrastive learning relates to graph representation learning, molecular representation learning, contrastive learning, and other domains. However, this work is a technical paper, not a literature review, and considering the limitation of the main text space, we only discuss the most relevant parts of the literature and leave more space for the technical details and experimental validations.
>
> As for the simple methods that perform on Tox21, please refer to the answer to Q2.2.
>
> **Q2.1:** The experimental part is limited and all methods operate at performances that are far below usual approaches to molecular property prediction. The authors only perform experiments for molecular property prediction on the so-called MoleculeNet benchmark. Since the method should generally work for graph pre-training, the authors should also perform experiments on graph that arise in a different domain (e.g. documents or social networks).
>
> **Answer:** In this paper, our motivation is to address the two issues in molecular contrastive learning, namely the molecular semantics alteration caused by data augmentation and the poor generalization capability with domain knowledge. Meanwhile, MoleculeNet is currently the most ubiquitous dataset for validating molecular contrastive learning models and has been adopted in many articles on molecular property prediction; thus, we are consistent with these methods and employ MoleculeNet for LGCL validation.
>
> Moreover, although LGCL is general for graph pre-training, the design concept (the addressed issues) may not fit other domains. First, as discussed in Section 4.1, a vertex with $e$ edges in $G$ will produce $e \times (e − 1)/2$ edges in $L(G)$, which could lead to severe runtime complexity when the original graphs are dense (i.e., social networks). Therefore, our method only suits sparse graphs. Please refer to Appendix B for the volumes of line graphs from social network domains.
>
> Second, the issues discussed in this paper may not exist or be slight in other domains. For example,  information is usually redundant in social networks, and data augmentation will not cause serious damage to semantic information. Please refer to Obs. 3 in GraphCL[1] with data augmentations. While in the text pre-training task, current research mainly focuses on attention mechanisms.
>
> [1] Yuning You, Tianlong Chen, Yongduo Sui, Ting Chen, Zhangyang Wang, and Yang Shen. Graph contrastive learning with augmentations. Advances in Neural Information Processing Systems, 33:5812–5823, 2020.
>
> **Q2.2:** Furthermore, the usual encoding or representation of molecular graphs are extended-connectivity fingerprints [2] that just encode subgraphs of different size. Using this simple and efficient representation, one can easily outperform the presented method using linear probing (see my table below). Thus, one can save all the computational costs that is used for pre-training the suggested encoder. The authors should include baselines into their table (such as linear probing on frequently used molecular descriptors), reference usual performances on these datasets (e.g. in MoleculeNet), and suggest a pre-training method which at least matches the performance of molecular descriptors.
>
> **Answer:** The results that Reviewer KhWt lists come from different experimental settings. Specifically, as shown in the **split** column, the ECFP adopts the `random` split for train/validation/test dataset generation. However, in our experimental setting, we employ the `scaffold` split that provides a more realistic estimate of model performance in prospective evaluation compared to random split. Please refer to Appendix E in Hu et al. ICLR 2020[2]. Moreover, the scaffold split provides more difficult prediction tasks and is also the mainstream data split method in the current molecular contrastive learning. For a fair comparison, we adopt this ubiquitous data split method in this paper.
>
> As for the effort devoted to pre-training, please refer to the article "Does GNN Pretraining Help Molecular Representation?" (NIPS'22) [3].
>
> [2] W Hu, B Liu, J Gomes, M Zitnik, P Liang, V Pande, and J Leskovec. Strategies for pre-training graph neural networks. International Conference on Learning Representations (ICLR), 2020.
>
> [3] Sun R. Does GNN Pretraining Help Molecular Representation?. Advances in Neural Information Processing Systems, 2022.

---

> ### Author Response · Authors · 2022-11-14
> **Initial Response to Reviewer KhWt (Part 2)**
>
> **Q3:** Lack of error bars, confidence intervals and statistical test for many performance metrics. The performance metrics in Figure 3 and Table 2 are provided withouth error bars or confidence intervals, such that differences could just arise by chance. The authors should perform repeated training runs, or perform cross-validation to obtain error bars.
>
> **Answer:** Thanks for your kind suggestions. Actually, the experimental setting of Figure 3 and Table 2 is in line with Table 1; put differently, we perform 10 different runs and report the average test ROC-AUC(%). Considering the brevity and readability, we omit the standard deviation. We have fixed this in the new revision.
>
> **Q4:** Unclear balancing strategies for the loss function and missing hyperparameter selection procedure. The authors use a combination of three loss functions to pre-train their GNN. However, such combination is usually hard to balance since the losses can be at completely different scales or move to different scales during learning -- a major problem, e.g., in deep reinforcment learning. Suprisingly, in this work the balancing parameters α and  β could just be set to 1. Generally, it is unclear how the many hyperparameters that come with this architecture (scale parameters, learning rate schedules, GNN architecture, regularization, ...) have been selected. The authors should describe their hyperparameter selection strategy and their metric how to select good hyperparameters.
>
> **Answer:** Thanks for noting this. In this work, we introduce two hyper-parameters, $\alpha$ and $\beta$, for loss balance in the pre-training stage. To ensure reproducibility and clearly show the essential effectiveness of the two proposed losses rather than the two hyper-parameters, we fix $\alpha$ and $\beta$ to 1. The other hyper-parameters in the pre-training phase are fixed and consistent with Hu et al. 2020[2], the standard architecture for molecular pre-training model testing.
>
> As also concerned by Reviewer awP4, we further inspected the hyper-parameter sensitivity of LGCL and tuned the candidates of $\alpha$ and $\beta$ in the range of [0.01, 0.1, 1, 10, 100], respectively. In the tunning of $\alpha$, we fix the $\beta$ to 1 and vice versa. Please refer to Appendix C.1 in the new revision and Answer to Q1.1 of Review awP4 for more details.
>
> As for the hyper-parameters regarding downstream tasks, despite those mentioned in **Fine-tuning details** (i.e., learning rate, batch size, dropout ratio, and graph pooling), the other hyper-parameters, such as learning rate schedules, GNN architecture, regularization and so on, are also fixed and consistent with GraphCL[1] and Hu et al. 2020[2] because our code is implemented based on the code repository of GraphCL (https://github.com/Shen-Lab/GraphCL). The tuned hyper-parameters are selected by the grid search on the validation sets.
>
> **Q5:** The proposed strategy to pre-train graph neural networks appears novel; differences and similarities to other pre-training strategies, e.g. [3], should be described clearer.
>
> **Answer:** Thank you for approving the novelty of our work. First, the mentioned article, "Structure-Preserving Graph Representation Learning", has **not been officially published**. More importantly, this work aims at the **semi-supervised node classification** task and excellent robustness under noise perturbation on graph structure or node features. However, the domain that our work focuses on is molecular contrastive learning which concentrates on the **global feature of molecules**. Given the method details, this article is similar to an article termed Geom-GCN (GEOM-GCN: GEOMETRIC GRAPH CONVOLUTIONAL NETWORKS, ICLR2020), in which they construct another connection layer outside the existing topological space.
>
> **Q6:** The code has not been provided, such that this work is hardly reproducible.
>
> **Answer:** As mentioned in the main text and noticed by Reviewer xTh3, we will release the source code after the acceptance.

---

> ### Comment · Reviewer_KhWt · 2022-12-12
> **Summary after rebuttal**
>
> I thank the authors for their rebuttal. Some minor things have been clarified, but my main concerns remain.
>
> ### Embedding in other methods and baselines has not improved.
> The authors do not improve on my suggested embedding into related works that are a bit outside of this particular community that focuses on pre-training GNNs. The authors mentioned that this is a technical paper and not a review paper, but it is basic scientific practice to embed one's own work into related works. However, there are communities that work on this particular topic, activity and property prediction, for decades and almost a decade with deep learning methods. This should not be ignored. Concretely, ref [2], Table 4, demonstrates that ECFP fingerprints with MLPs perform better than almost all GNN pre-training methods and this method should, aside from other methods, be used as a baseline in all comparisons.
>
> ### Loss terms and balancing.
> There are hardly any contrastive learning methods or pre-training strategies in other domains (aside from RL), that use more than two loss terms. There must be an exceptional reason why three loss terms should be necessary. Based on the ablation study, the differences between using additional loss terms or not does not change much: the performances are almost the same, or within error bars. It is still unclear why the loss terms are necessary.
>
> ### Lack of clarity about hyperparameters and hyperparameter selection.
> The authors now mention a validation set, but it remains unclear how the validation set was used, how large it is, and how the hyperparameters were concretely selected: is this done via nested cross-validation or just on train/val/test? Grid search is only mentioned for fine-tuning. Is this done for each of the downstream datasets differently or the same? In C.1 it appears that the authors would pick the best hyperparameters on the test set.
>
>
> References:
> [1] Jiang, D., Wu, Z., Hsieh, C. Y., Chen, G., Liao, B., Wang, Z., ... & Hou, T. (2021). Could graph neural networks learn better molecular representation for drug discovery? A comparison study of descriptor-based and graph-based models. Journal of cheminformatics, 13(1), 1-23.
> [2] Honda, S., Shi, S., & Ueda, H. R. (2019). Smiles transformer: Pre-trained molecular fingerprint for low data drug discovery. arXiv preprint arXiv:1911.04738.

---

> > ### Author Response · Authors · 2022-12-13
> > **Thanks for the follow up (part 2)**
> >
> > **Q3:** Lack of clarity about hyperparameters and hyperparameter selection.
> >
> > > The authors now mention a validation set, but it remains unclear how the validation set was used, how large it is, and how the hyperparameters were concretely selected: is this done via nested cross-validation or just on train/val/test? Grid search is only mentioned for fine-tuning. Is this done for each of the downstream datasets differently or the same? In C.1 it appears that the authors would pick the best hyperparameters on the test set.
> >
> > **Answer:** As described in the main text, we follow the experimental setup under the guidance of Hu et al. (2020) [5]. More details about the dataset split are also attached in the Appendix A, in which the **Dataset splitting** shows that "**The split for train/validation/test sets is 80%:10%:10%**".
> >
> > The hyper-parameters for all downstream tasks' fine-tuning are grid-searched based on the performance of validation set. To avoid misunderstanding, we attach the results of validation set regarding the sensitivity analysis of $\alpha$ and $\beta$:
> >
> > $\alpha$ | 0.01 | 0.1 | 1 | 10 | 100 |
> > :---:|:---:|:---:|:---:|:---:|:---:
> > BACE |83.92$\pm$1.62|**87.78$\pm$1.35**|85.08$\pm$0.73|83.34$\pm$2.02|82.78$\pm$1.77
> > BACE-val| 70.39$\pm$1.06 | 75.62$\pm$1.09 | 74.15$\pm$1.46 |  70.90$\pm$0.54 |  69.92$\pm$0.98
> > SIDER|62.56$\pm$0.55|63.18$\pm$0.73|63.02$\pm$0.75|62.15$\pm$0.92|**64.09$\pm$0.45**
> > SIDER-val | 62.96$\pm$0.57 |  63.27$\pm$0.84 |  62.61$\pm$0.19 |  62.57$\pm$0.43 |  64.42$\pm$1.05
> > ClinTox|**78.11$\pm$3.79**|77.84$\pm$2.12|74.95$\pm$4.09|76.99$\pm$2.65|75.92$\pm$0.49
> > ClinTox-val | 90.16$\pm$1.72 |   87.41$\pm$0.85 |   84.31$\pm$2.61 |   89.70$\pm$1.39 |   85.65$\pm$0.74
> >
> > $\beta$  | 0.01 | 0.1 | 1 | 10 | 100
> > :---:|:---:|:---:|:---:|:---:|:---:
> > BACE |83.80$\pm$1.25|80.96$\pm$1.12|**85.08$\pm$0.73**|83.04$\pm$1.63|82.34$\pm$2.21
> > BACE-val| 70.86$\pm$1.37 |  69.62$\pm$1.08 |  74.15$\pm$1.46 |  73.36$\pm$1.58 |  71.19$\pm$0.95
> > SIDER|63.12$\pm$0.79|63.32$\pm$0.60|63.02$\pm$0.75|**63.65$\pm$1.56**|63.01$\pm$1.45
> > SIDER-val | 63.01$\pm$0.58 |   63.59$\pm$1.01 |   62.61$\pm$0.19 |   63.74$\pm$0.41 |   62.51$\pm$0.54
> > ClinTox|74.63$\pm$2.42|78.15$\pm$4.09|74.95$\pm$4.09|**85.88$\pm$3.16**|78.90$\pm$2.20
> > ClinTox-val | 86.08$\pm$2.74 |  88.21$\pm$1.16 |  84.31$\pm$2.61 |  91.97$\pm$2.24 |  89.35$\pm$1.52
> >
> >
> > [1] Jiang, D., Wu, Z., Hsieh, C. Y., Chen, G., Liao, B., Wang, Z., ... & Hou, T. (2021). Could graph neural networks learn better molecular representation for drug discovery? A comparison study of descriptor-based and graph-based models. Journal of cheminformatics, 13(1), 1-23.
> >
> > [2] Honda, S., Shi, S., & Ueda, H. R. (2019). Smiles transformer: Pre-trained molecular fingerprint for low data drug discovery. arXiv preprint arXiv:1911.04738.
> >
> > [3] Shengchao Liu, Hanchen Wang, Weiyang Liu, Joan Lasenby, Hongyu Guo, and Jian Tang. Pre-training molecular graph representation with 3d geometry. In International Conference on Learn- ing Representations, 2022.
> >
> > [4] Dongki Kim, Jinheon Baek, and Sung Ju Hwang. Graph self-supervised learning with accurate discrepancy learning. Advances in Neural Information Processing Systems, 2022.
> >
> > [5] W Hu, B Liu, J Gomes, M Zitnik, P Liang, V Pande, and J Leskovec. Strategies for pre-training graph neural networks. International Conference on Learning Representations (ICLR), 2020.

---

> > ### Author Response · Authors · 2022-12-13
> > **Thanks for the follow up (part 1)**
> >
> > **Q1:** Embedding in other methods and baselines has not improved.
> >
> > > The authors do not improve on my suggested embedding into related works that are a bit outside of this particular community that focuses on pre-training GNNs. The authors mentioned that this is a technical paper and not a review paper, but it is basic scientific practice to embed one's own work into related works. However, there are communities that work on this particular topic, activity and property prediction, for decades and almost a decade with deep learning methods. This should not be ignored. Concretely, ref [2], Table 4, demonstrates that ECFP fingerprints with MLPs perform better than almost all GNN pre-training methods and this method should, aside from other methods, be used as a baseline in all comparisons.
> >
> > **Answer:** Thanks for your kind suggestion. The first suggested reference, entitled "**Could graph neural networks learn better molecular representation for drug discovery? A comparison study of descriptor-based and graph-based models**" and published in "**Journal of cheminformatics**", will be cited with the acceptance of this work. The second suggested reference, entitled "**Smiles transformer: Pre-trained molecular fingerprint for low data drug discovery**", has not been officially published. I think you will agree that it is not professional at citing unpublished works.
> >
> > As for the results of ECFP in Table 4 of ref [2], first, this work [2] has not been officially published. Second, the results of ECFP are produced via **three** independent runs without the error bars. The results of LGCL are obtained over **ten** different runs with different seeds, and the seeds are set as 0 to 9. In summary, there does not exist a direct comparison between LGCL and the results of ECFP from unpublished work because of different experiment settings.
> >
> > In this work, we focus on a novel domain of molecular pre-training, molecular contrastive learning. In light of the results in Table 1 of the main text and results after tuning $\alpha$ and $\beta$, LGCL outperforms the previous works regarding molecular contrastive learning.
> >
> >
> > **Q2:** Loss terms and balancing.
> >
> > > There are hardly any contrastive learning methods or pre-training strategies in other domains (aside from RL), that use more than two loss terms. There must be an exceptional reason why three loss terms should be necessary. Based on the ablation study, the differences between using additional loss terms or not does not change much: the performances are almost the same, or within error bars. It is still unclear why the loss terms are necessary.
> >
> > **Answer:** In this work, we focus on molecular contrastive learning. Among the selected baselines regarding molecular contrastive learning, except for GraphCL, other works have more than one loss. In particular, **GraphMVP[3] and D-SLA[4] have three loss terms**. Therefore, more than two loss terms may be a common and effective method in molecular contrastive learning. As for the exceptional reason, intra-local contrastive loss is designed to alleviate over-smoothing along with line graph, and inter-local contrastive loss aims to enhance the contrastive learning from the node angle.
> >
> > Besides the technique reasons mentioned above, more practically, in the ablation study, the average result of LGCL among eight benchmarks without the two loss terms is 72.68%, which is lower than the current SOTA methods, such as D-SLA (74.14%) and RGCL (73.16%).

---

### Official Review · Reviewer_TRA8 · 2022-10-26

**Confidence:** 4
**Correctness:** 3
**Technical Novelty And Significance:** 1
**Empirical Novelty And Significance:** 3
**Recommendation:** 5

**Clarity, Quality, Novelty And Reproducibility:**

Clarity: Good, Easy to follow the paper.
Quality: Poor.
Novelty: not novel.
Reproducibility: Not sure.

**Strength And Weaknesses:**

**Strength**
Paper is well writen and easy to follow the idea

**Weakness**
Overall, the idea of this paper is not novel and the method propsed by the authors is incremental.
1. The idea of introducing the line graph of the molecuars to graph representation learning is trivial. It has been explorer in the various GNN papers. Although this paper might be the first one to apply the line graph to the contastive learning, the effort has been made is minor and not significant. It just apply line graph as a contrastive veiws.

2. There is no evidence to show that the proposed intra-local contrastive loss can allivate the oversmooth problem, however, the author claim for that. Need either thertical analysis or empericial result to show this.

3. Some reuslt is much worse than the baseline, can author give some explaination on that.

**Summary Of The Paper:**

This paper fouse on learning good representation for molecuars. The authors introudce line graph to graph contrastive learning and propose two contrastive loss (i.e., inter contrastive loss and intra contrastive loss) to retain molecular semantics. Experiment shows proposed LGCL ahieve SOTA performance on several datasets.

**Summary Of The Review:**

The author give a trival solution to the learning the represeatation for molecuars. The novelty of this paper is not enough for ICLR.

---

> ### Author Response · Authors · 2022-11-14
> **Initial Response to Reviewer TRA8**
>
> **Q1:** The idea of introducing the line graph of the molecuars to graph representation learning is trivial. It has been explorer in the various GNN papers. Although this paper might be the first one to apply the line graph to the contastive learning, the effort has been made is minor and not significant. It just apply line graph as a contrastive veiws.
>
> **Answer:** Reviewer TRA8 has a severe misunderstanding of this article. In this paper, our motivation is not to apply the line graph as a contrastive view but to address the two issues in molecular contrastive learning, namely the molecular semantics alteration caused by data augmentation and the poor generalization capability with domain knowledge.
> Therefore, our contribution in this article goes beyond introducing line graphs to molecular contrastive learning. More importantly, as we conclude at the end of the introduction, we give an excellent solution for preserving the full semantic structure in molecular contrastive learning without relying on domain knowledge.
>
> Furthermore, the introduction of line graphs is also because line graphs have achieved excellent performance in many tasks regarding graph representation learning, which exactly inspired us to apply line graphs to graph contrastive learning. Of course, as you also noticed, this work is the first study to adopt line graphs for molecular contrastive learning. Meanwhile, as discussed in our line graph transformation, simply applying the line graph as a contrastive view still has issues, namely information inconsistency and over-smoothing, which need to be addressed. If only the line graph is introduced, as shown in the ablation study, although it is effective, the performance is far inferior to the current molecular contrastive learning model. Therefore, it is necessary to further solve the problems along with the line graph so that LGCL can fully exhibit its excellent representation capability.
>
> **Q2:** There is no evidence to show that the proposed intra-local contrastive loss can allivate the oversmooth problem, however, the author claim for that. Need either thertical analysis or empericial result to show this.
>
> **Answer:** Thanks for your kind suggestion. In the ablation study, we empirically inspect the effect of intra-local contrastive loss in over-smoothing alleviating. Please refer to the ***The effect of intra-local contrastive loss*** in the main text.
>
> As suggested, a visualization case study is presented to further directly show how the intra-local contrastive works. Specifically, we calculate the similarities among nodes within the same graph from the pre-training dataset via the model after 100-epoch pre-training. Please refer to **Appendix C.3** in the new revision for details. In summary, the introduction of line graph leads to a right shift of node similarity distribution, that is, over-smoothing. Then, a left shift on node similarity distribution achieved with the introduction of the proposed local loss suggests its effectiveness in over-smoothing alleviating.
>
> **Q3:** Some reuslt is much worse than the baseline, can author give some explaination on that.
>
> **Answer:** Thanks for noting this. First of all, LGCL achieves SOTA performance on 6 out of 8 benchmarks, the highest average ROC-AUC, and the highest average ranking position, which suggests the effectiveness of LGCL. The unsatisfactory results only showed on the BBBP and ClinTox. One reason could be the difference between the message-passing scheme of the line graph encoder and the graph encoder. The line graph encoder transfers the edge features, while the graph encoder transfers the node feature.
> Among the various molecular features, BBBP and ClinTox may be more reliable on the node feature, which also implies the future research direction.
>
>
> It is worth noting that, as suggested by Reviewer awP4, superior results can be obtained by the tunning of $\alpha$ and $\beta$ in the pre-training phase. The updated results of LGCL on BACE, SIDER, and ClinTox are shown below. We will also update the main text while finishing the sensitivity experiments. Please refer to the answer to Q1.1 of Review awP4 for more details.
>
>   -| BACE | SIDER | ClinTox
> :---:|:---:|:---:|:---:
> LGCL in Main Text |84.68$\pm$0.73|63.37$\pm$0.56|77.59$\pm$1.54
> LGCL after Sensitivity |87.78$\pm$1.35($\uparrow$**3.1**)|64.09$\pm$0.45($\uparrow$**0.72**)|85.88$\pm$3.16($\uparrow$**8.29**)

---

> > ### Comment · Reviewer_TRA8 · 2022-11-18
> > **Thanks for the response**
> >
> > 1) The author provides additional empirical results to show their method is better. However, for publishing in ICLR, I still feel it lacks some theoretical understanding of how the proposed method helps the GCL in learning better representation.
> >
> > 2) If more semantic information was introduced by a line graph, the remaining problem was the so-called information inconsistency and over-smoothing, as the author indicates. So, why proposed two CL loss can help typical CL loss to gain more consistency between two views? Will it improve the alignment or the uniformity of the embedding?
> >
> > To my knowledge, there are many ways to alleviate over-smoothing (e.g., reduce the GNN layer, jumping connection), as the purpose of this work is to tackle the over-smoothing problem these simple methods should be discussed
> >
> > Overall, the response solves part of my concerns. So I raise my score from 3 to 5.
> >
> > The author may provide more explanation and I am willing to increase my score further.

---

> > > ### Author Response · Authors · 2022-11-19
> > > **Thanks for the follow up**
> > >
> > > **Q1:**  The author provides additional empirical results to show their method is better. However, for publishing in ICLR, I still feel it lacks some theoretical understanding of how the proposed method helps the GCL in learning better representation.
> > >
> > > **Answer:** Thanks for noting this. Further theoretical understanding of how LGCL obtains better performance is discussed in the new revision. Please refer to **Appendix E** for details.
> > >
> > > In summary, Hassler Whitney (1932) [1] proved that the structure of a connected graph $G$ can be recovered completely from its line graph; thus, we can have $I(L(G), Y) = I(G, Y)$. However, according to the data processing inequality [2], $I(t(G), Y) \le I(G, Y)$, in which the function $t$ is a general data augmentation function, such as node dropping and edge perturbation. Therefore, we have $I(L(G), Y) \ge I(t(G), Y)$, which suggests that the line graph contains more information than data augmentation for target prediction.
> > >
> > > [1] Hassler Whitney. Congruent graphs and the connectivity of graphs. American Journal of Mathematics, 54(1):150–168, 1932.
> > >
> > > [2] Cover, Thomas M. Elements of information theory. John Wiley & Sons, 1999.
> > >
> > >
> > > **Q2:** If more semantic information was introduced by a line graph, the remaining problem was the so-called information inconsistency and over-smoothing, as the author indicates. So, why proposed two CL loss can help typical CL loss to gain more consistency between two views? Will it improve the alignment or the uniformity of the embedding?
> > >
> > > **Answer:** Thanks for noting this. Actually, the line graphs do not introduce more semantic information but avoid the loss of semantic information caused by data augmentation in previous molecular contrastive learning methods. Moreover, information consistency is defined between two views, while over-smoothing is internal to the view. Therefore, the proposed edge attribute fusion aims to ensure the consistency of information between views, while the intra-local contrastive loss is designed for over-smoothing.
> > >
> > > In the previous molecular contrastive learning methods, the data augmentation damaged the corresponding relationship between views; thus, it is hard to transfer information between views. The line graph retains this relationship so that we can enhance the consistency of information between views through effective information exchange.
> > >
> > > In the ablation study, we empirically validate the effectiveness of edge attribute fusion in information inconsistency alleviating. Please refer to the **The effect of edge attribute fusion** in the main text.
> > > Further visualization case studies are presented to directly show how the edge attribute fusion works. Please refer to **Appendix C.4** in the new revision for details. All in all, the introduction of edge attribute fusion leads to a right shift of view similarity distribution of positive pairs and a faster and lower convergence curve of NT-Xent loss. Therefore, edge attribute fusion can improve the alignment or the uniformity of the embedding.
> > >
> > > **Q3:** To my knowledge, there are many ways to alleviate over-smoothing (e.g., reduce the GNN layer, jumping connection), as the purpose of this work is to tackle the over-smoothing problem these simple methods should be discussed.
> > >
> > > **Answer:** Thanks for noting this. To validate the effectiveness of LGCL, we stick to the evaluation protocols of Hu et al. (2020) [3], in which the graph encoder is designed with fixed 5 GNN layers and without the jump connection. Therefore, we do not employ these methods (e.g., reduce the GNN layer, jumping connection) for a fair comparison.
> > >
> > > Again, the purpose of this work is to address the two issues in molecular contrastive learning, namely the molecular semantics alteration caused by data augmentation and the poor generalization capability with domain knowledge. The over-smoothing is an accompanying issue with the line graph.
> > >
> > > [3] W Hu, B Liu, J Gomes, M Zitnik, P Liang, V Pande, and J Leskovec. Strategies for pre-training graph neural networks. In ICLR, 2020.

---

### Official Review · Reviewer_awP4 · 2022-10-26

**Confidence:** 4
**Correctness:** 3
**Technical Novelty And Significance:** 2
**Empirical Novelty And Significance:** 3
**Recommendation:** 6

**Clarity, Quality, Novelty And Reproducibility:**

The reproducibility is good because of the promise of releasing original code and detailed experimental results.
The figures and writing are clear and easy to follow.
The quality of the paper is roughly satisfactory and exceeds the minimum threshold for top conferences.
The technical novelty is limited.

**Strength And Weaknesses:**

Strength:
1. The paper is well-written and easy to follow. The storyline is clear, and the figures are intuitive.

2. The authors are willing to share the code link after acceptance. The reproducibility of this paper is satisfactory.

3. The experimental setting is very exhaustive. The authors describe almost all the interesting experimental details.

Weakness:
1. The experimental settings are kind of solid but not enough. The experimental section only includes overall performance comparison and ablation study in two different settings. The hyper-parameter sensitivity experiments are missing. And the authors report that the pretraining experiments of their model require 20 hours to converge. I am curious about the efficiency of their method compared with other methods.

2. Some model designs are too simple to come up with. The idea behind the model is kind of interesting. But some model designs are too simple to come up with. For example, the GNN architecture and contrastive loss are similar to other works.

3. The authors claim that edge attribute fusion is essential to fix the inconsistency problem between the representations from two views. But the experimental results shown in figure 3 demonstrate that the model performs worse on half of all the datasets with this attribute fusion.

**Summary Of The Paper:**

The paper claims that random or learnable graph augmentation methods may inevitably vary molecular semantics and make them inappropriate applied to molecular structure. The authors propose a novel approach to augment the original graph, i.e., converting it to a line graph. The edge attribute fusion tackles the inconsistent representation issue. Two following contrastive losses enhance representation learning.

**Summary Of The Review:**

The paper would benefit from several revisions before publication, including adding some experiments, improving the model design, and checking the necessity of the components carefully.

---

> ### Author Response · Authors · 2022-11-14
> **Initial Response to Reviewer awP4 (Part 1)**
>
> We sincerely thank you for your constructive and helpful comments. We initially address all your concerns below.
>
> **Q1.1:** The experimental settings are kind of solid but not enough. The experimental section only includes overall performance comparison and ablation study in two different settings. The hyper-parameter sensitivity experiments are missing.
>
> **Answer:** Thanks for your suggestion. Besides the experiment under the setting of transfer learning, we further evaluate LGCL via unsupervised and semi-supervised learning in Appendix B and obtain superior performance. In hyper-parameter tunning, besides the general parameters (i.e., learning rate, batch size, dropout ratio, etc.), we introduce two hyper-parameters, $\alpha$ and $\beta$, for loss balance in pre-training stage. To clearly show the essential effectiveness of the two losses rather than the two hyper-parameters, we fix $\alpha$ and $\beta$ to 1. The other hyper-parameters in pre-training phase are also fixed and consistent with Hu. et al. 2020[1].
>
> To further inspect the hyper-parameter sensitivity of LGCL, we tuned the candidates of $\alpha$ and $\beta$ in the range of [0.01, 0.1, 1, 10, 100], respectively. In the tunning of $\alpha$, we fix the $\beta$ to 1 and vice versa. Please refer to **Appendix C.1** in the new revision for more details. Considering the time limit, here, we only give the results of BACE, SIDER and ClinTox in the tunning process. More results will be updated with the acceptance.
>
>
> $\alpha$ | 0.01 | 0.1 | 1 | 10 | 100 |
> :---:|:---:|:---:|:---:|:---:|:---:
> BACE |83.92$\pm$1.62|**87.78$\pm$1.35**|85.08$\pm$0.73|83.34$\pm$2.02|82.78$\pm$1.77
> SIDER|62.56$\pm$0.55|63.18$\pm$0.73|63.02$\pm$0.75|62.15$\pm$0.92|**64.09$\pm$0.45**
> ClinTox|**78.11$\pm$3.79**|77.84$\pm$2.12|74.95$\pm$4.09|76.99$\pm$2.65|75.92$\pm$0.49
>
> $\beta$  | 0.01 | 0.1 | 1 | 10 | 100
> :---:|:---:|:---:|:---:|:---:|:---:
> BACE |83.80$\pm$1.25|80.96$\pm$1.12|**85.08$\pm$0.73**|83.04$\pm$1.63|82.34$\pm$2.21
> SIDER|63.12$\pm$0.79|**63.32$\pm$0.60**|63.02$\pm$0.75|63.65$\pm$1.56|63.01$\pm$1.45
> ClinTox|74.63$\pm$2.42|78.15$\pm$4.09|74.95$\pm$4.09|**85.88$\pm$3.16**|78.90$\pm$2.20
>
>
> It is worth noting that superior results can be obtained from tuning the two hyper-parameters. The updated results of LGCL on BACE, SIDER, and ClinTox are shown below. We will also update the main text while finishing the sensitivity experiments.
>
>  -| BACE | SIDER | ClinTox
> :---:|:---:|:---:|:---:
> LGCL in Main Text |84.68$\pm$0.73|63.37$\pm$0.56|77.59$\pm$1.54
> LGCL after Sensitivity |87.78$\pm$1.35($\uparrow$**3.1**)|64.09$\pm$0.45($\uparrow$**0.72**)|85.88$\pm$3.16($\uparrow$**8.29**)
>
>
> [1] W Hu, B Liu, J Gomes, M Zitnik, P Liang, V Pande, and J Leskovec. Strategies for pre-training graph neural networks. International Conference on Learning Representations (ICLR), 2020.
>
> **Q1.2:** And the authors report that the pretraining experiments of their model require 20 hours to converge. I am curious about the efficiency of their method compared with other methods.
>
> **Answer:** Thanks for the kind suggestion. We rerun baselines and report their pre-training time. Detailed time consumptions of baselines and LGCL with the introduction of each proposed component are discussed in the new revision. Please refer to **Appendix C.2** in the new revision for model efficiency. In summary, because the contrastive views of LGCL are static, the time required by LGCL for 2 million molecular graphs from ZINC15 pre-training (100 epochs) is much less than the time needed by baselines except AutoGCL. Here, we give a brief table for training time comparison under the same setting in our experimental environment. Note that we set the `num_workers` in `dataloader` to `default` for a fair comparison.
> In particular, LP-Info[2] requires too much time to finish its pre-training stage; thus, we only run one epoch and report its time consumption.
>
>  2M ZINC15 | GraphCL | JOAOv2 | LP-Info | AD-GCL | AutoGCL | RGCL | LGCL
> :---:|:---:|:---:|:---:|:---:|:---:|:---:|:---:
> Hours (100 epochs) |79.99|162.20|15h50min (1 epoch)|46.88|22.15|111.99| 31.49
>
>  2M ZINC15 | LGCL-w/o AF L\_IntraC L_InterC | LGCL-w/o L\_IntraC L_InterC | LGCL-w/o  L_InterC  | LGCL
> :---:|:---:|:---:|:---:|:---:
> Hours (100 epochs) |26.15|26.37|30.71|31.49
>
> [2] Yuning You, Tianlong Chen, Zhangyang Wang, and Yang Shen. Bringing your own view: Graph contrastive learning without prefabricated data augmentations. WSDM ’22, pp. 1300–1309, New York, NY, USA, 2022. Association for Computing Machinery.

---

> ### Author Response · Authors · 2022-11-14
> **Initial Response to Reviewer awP4 (Part 2)**
>
> **Q2:** Some model designs are too simple to come up with. The idea behind the model is kind of interesting. But some model designs are too simple to come up with. For example, the GNN architecture and contrastive loss are similar to other works.
>
> **Answer:** In this paper, our motivation is to address the issues of molecular semantics alteration and generalization capability in molecular contrastive learning. Therefore, if we can solve the two problems through a simple method, it should be a better choice than those complex methods. Moreover, the design of LGCL is not simple. Specifically, besides the introduction of the line graph, to further address the issues along with the line graph (i.e., information inconsistency and over-smoothing), we propose the edge attribute fusion and two local losses.
>
> Furthermore, the two issues belong to the realm of molecular contrastive learning; thus, the entire framework is still maintained within the twin-tower architecture of graph contrastive learning, referring to the design of GraphCL. However, our model has unique innovations, such as edge attribute fusion and two local losses, which cannot be achieved by previous methods due to the lack of one-to-one correspondence between the two contrastive views from data augmentation.
>
> **Q3:** The authors claim that edge attribute fusion is essential to fix the inconsistency problem between the representations from two views. But the experimental results shown in figure 3 demonstrate that the model performs worse on half of all the datasets with this attribute fusion.
>
> **Answer:** Thanks for noting this. There is indeed a performance deterioration in Figure 3 with the introduction of attribute fusion, but it only exists in three datasets (i.e., SIDER, ClinTox, and HIV). Moreover, the results in Figure 3 come from the model without pre-training, which implies the other issue, over-smoothing, along with the line graph, because we do not introduce the local loss to the downstream tasks. Comparing the results between the 1st and 5th row in Table 2, which shows the results after pre-training, we can find an overall performance improvement except for a negligible performance degradation on Tox21($\downarrow$0.03%). If comparing the results between the 4th and 6th row (with the loss for over-smoothing) in Table 2, all performances improve.
>
> Again, we appreciate Reviewer awP4 for the constructive and helpful comments that help up to improve the model performance and the paper quality. If there are further concerns, please do let us know. We would be happy to address them.

---

### Author Response · Authors · 2022-11-25
**General response to ACs and Reviewers**

We appreciate the effort and support from all the reviewers. Based on the suggestion of Reviewer awP4, we conduct the sensitivity experiments on the hyper-parameters for loss control, i.e., $\alpha$ and $\beta$. Now, we have finished the last experiment on MUV. Here, we update the results of LGCL. Specifically, we denote the results in the main text with $\alpha=1$ and $\beta=1$ as ***LGCL-Fix***, the results after hyper-parameter tunning as ***LGCL-Opt***. The results of LGCL-Fix and LGCL-Opt are attached below:

-| BBBP |  Tox21 | ToxCast | SIDER | ClinTox | MUV | HIV | BACE | Avg.
:---:|:---:|:---:|:---:|:---:|:---:|:---:|:---:|:---:|:---:
LGCL-Fix|70.99$\pm$1.05 | 76.95$\pm$0.43 | 64.71$\pm$0.72 | 63.37$\pm$0.56 | 77.59$\pm$1.54 | 77.70$\pm$3.00 | 78.69$\pm$1.10 | 84.68$\pm$0.73 | 74.33
LGCL-Opt| 73.11$\pm$1.08 | 77.45$\pm$0.57 | 65.07$\pm$0.52 |64.09$\pm$0.45 |85.88$\pm$3.16 |79.42$\pm$4.26 |79.06$\pm$1.22|87.78$\pm$1.35 |76.48($\uparrow$2.15)

---

### Decision · Program_Chairs · 2023-01-20

**Decision:**

Reject

**Justification For Why Not Higher Score:**

After rebuttal and AC-reviewer meeting, a number of concerns remained as explained above.

**Justification For Why Not Lower Score:**

N/A

**Metareview: Summary, Strengths And Weaknesses:**

This paper has been evaluated by five reviewers who scored the paper as 3,5,6,6,8. Reviewers liked the paper but also had a number of concerns.

Firstly, they had concerns about the lack of strong motivation for Eq. 10 and 11, or the use of the line graph. Reviewers acknowledged the line graph is popular in molecular graphs but also noted that it was an arbitrary choice with a limited theoretical support why it would be the best choice for the second view of the original graph. It was not clear if other choices for forming views are really as problematic as authors claimed. One of reviewers had a concern also about inferior results compared with ECFP+linear probing and ECFP+MLP based on results from Honda, S., Shi, S., & Ueda, H. R. (2019). Smiles transformer: Pre-trained molecular fingerprint for low data drug discovery. arXiv preprint arXiv:1911.04738. From minor issues, some reviewers noted they had concerns about the statistical significance of results in Table 1 as well as inconsistency of protocols which should have been indicated for clarity: (random/scaffold) and the validation procedure (train/val/test or k-fold cross-validation).

The above findings were discussed during the AC-reviewers meeting.

On balance, the paper requires another round of reviews and is not ready for publication.

**Summary Of Ac-Reviewer Meeting:**

During the meeting, concerns about the lack of strong motivation for Eq. 10 and 11, or the use of the line graph were discussed. Reviewers acknowledged the line graph is popular in molecular graphs but also noted that it was an arbitrary choice with a limited theoretical support why it would be the best choice for the second view of the original graph. It was not clear if other choices for forming views are really as problematic as authors claimed. One of reviewers had a concern also about inferior results compared with ECFP+linear probing and ECFP+MLP based on results from Honda, S., Shi, S., & Ueda, H. R. (2019). Smiles transformer: Pre-trained molecular fingerprint for low data drug discovery. arXiv preprint arXiv:1911.04738. From minor issues, some reviewers noted they had concerns about the statistical significance of results in Table 1 as well as inconsistency of protocols which should have been indicated for clarity: (random/scaffold) and the validation procedure (train/val/test or k-fold cross-validation). Additional aspects such as ablations were discussed and the experimental results. The meeting was a mix of in-person attendees and reviewers who messaged in their thoughts directly.